# Allelic composition of carotenoid metabolic genes in 13 founders influences carotenoid composition in juice sac tissues of fruits among Japanese citrus breeding population

Hiroshi Fujii[1☯], Keisuke Nonaka[1☯], Mai F. Minamikawa[2☯], Tomoko Endo[1☯], Aiko Sugiyama[3☯], Kosuke Hamazaki[2☯], Hiroyoshi Iwata[2☯], Mitsuo Omura[3☯], Takehiko Shimada[1☯]*

1 National Agriculture and Food Research Organization Institute of Fruit and Tea Tree Science, Shimizu, Shizuoka, Japan, 2 Laboratory of Biometry and Bioinformatics, Department of Agricultural and Environmental Biology, Graduate School of Agricultural and Life Sciences, The University of Tokyo, Bunkyo, Tokyo, Japan, 3 Faculty of Agriculture, Shizuoka University, Suruga, Shizuoka, Japan

☯ These authors contributed equally to this work.
* tshimada@affrc.go.jp

**Data Availability Statement:** All relevant data are within the manuscript and its Supporting Information files.

## Abstract

To enrich carotenoids, especially β-cryptoxanthin, in juice sac tissues of fruits via molecular breeding in citrus, allele mining was utilized to dissect allelic variation of carotenoid metabolic genes and identify an optimum allele on the target loci characterized by expression quantitative trait (eQTL) analysis. SNPs of target carotenoid metabolic genes in 13 founders of the Japanese citrus breeding population were explored using the SureSelect target enrichment method. An independent allele was determined based on the presence or absence of reliable SNPs, using trio analysis to confirm inheritability between parent and off-spring. Among the 13 founders, there were 7 *PSY* alleles, 7 *HYb* alleles, 11 *ZEP* alleles, 5 *NCED* alleles, and 4 alleles for the eQTL that control the transcription levels of *PDS* and *ZDS* among the ancestral species, indicating that some founders acquired those alleles from them. The carotenoid composition data of 263 breeding pedigrees in juice sac tissues revealed that the phenotypic variance of carotenoid composition was similar to that in the 13 founders, whereas the mean of total carotenoid content increased. This increase in total carotenoid content correlated with the increase in either or both β-cryptoxanthin and violaxanthin in juice sac tissues. Bayesian statistical analysis between allelic composition of target genes and carotenoid composition in 263 breeding pedigrees indicated that *PSY-a* and *ZEP-e* alleles at *PSY* and *ZEP* loci had strong positive effects on increasing the total carotenoid content, including β-cryptoxanthin and violaxanthin, in juice sac tissues. Moreover, the pyramiding of these alleles also increased the β-cryptoxanthin content. Interestingly, the offset interaction between the alleles with increasing and decreasing effects on carotenoid content and the epistatic interaction among carotenoid metabolic genes were observed and these interactions complexed carotenoid profiles in breeding population. These results revealed that allele composition would highly influence the carotenoid composition in citrus fruits. The allelic genotype information for the examined carotenoid metabolic genes in

**Funding:** This work was partially supported by a grant from the Ministry of Agriculture, Forestry, and Fisheries of Japan (Genomics-based Technology for Agricultural Improvement, HOR-2003, DNA-marker breeding project), by a grant from the Project of the Bio-oriented Technology Research Advancement Institution, NARO (the special scheme project on advanced research and development for next-generation technology, 150), and by a Grant-in-Aid for Scientific Research of Japan Society for the Promotion of Science (JSPS), Grant Number 23580055.

**Competing interests:** The authors have declared that no competing interests exist.

major citrus varieties and the trio-tagged SNPs to discriminate the optimum alleles (*PSY-a* and *ZEP-e*) from the rest would promise citrus breeders carotenoid enrichment in fruit via molecular breeding.

## Introduction

Carotenoids are lipophilic isoprenoid pigments biosynthesized from 5-carbon isoprene units and $C_{40}$ carotenoids and their derived $C_{30}$ apocarotenoids are the most abundant in nature. Most photosynthetic organisms produce carotenoids, which are essential for the survival of both plants and animals [1]. Carotenoids are involved in various functions in plants, including phyto-hormone precursor action [2] and environmental adaptation through the modulation of the photosynthetic apparatus [3]. Approximately 115 different carotenoids have been reported in citrus fruits, wherein the color of the fruit and peel are caused by carotenoid accumulation [4], and carotenoid content and composition vary among citrus varieties. For example, Satsuma mandarin (*Citrus unshiu* Marc.) and Ponkan mandarin (*C. reticulata* Blanco) predominantly accumulate β-cryptoxanthin in the juice sacs (1.5 mg/100FWG and 0.9 mg/100FWG, respectively), which are the major sources of β-cryptoxanthin in nature [5]. In contrast, Valencia orange (*C. sinensis* Osbeck) mainly accumulates violaxanthin isomers with 9-*cis*-violaxanthin (0.9 mg/100FWG) as the principal carotenoid. Lisbon lemon (*C. limon* Burm. f.) also accumulates β-cryptoxanthin as the principal carotenoid, but at a much lower level than Satsuma mandarin and Valencia orange. Epidemiologic studies have suggested that dietary intake of β-cryptoxanthin reduces the risks of eye diseases, certain cancers, osteoporosis and inflammation [6–9]. Therefore, in the Japanese citrus breeding program, the enrichment of carotenoids with health-promoting properties, especially β-cryptoxanthin, is an important objective with the aim of expanding citrus fruit consumption. Till date, our research institute has released β-cryptoxanthin enrichment varieties obtained by conventional cross breeding, such as 'Seinanohikari' (2.3mg/100gFWG) and 'Tsunokagayaki' (2.3mg/100gFWG) and 'Tamami' (2.0mg/100gFWG). The total carotenoid content and β-cryptoxanthin content of their parent varieties are not higher than those varieties, indicating a complicated regulation system involved in controlling the accumulation and composition of carotenoids by plural genetic loci in citrus fruits. The carotenoid composition varies among citrus varieties rather than species, and the carotenoid diversity in cultivated citrus is highly influenced by genetic factors [10]. Numerous carotenoid metabolic genes have been physiologically characterized at the molecular level [11–13], including several transcription factors, such as *CubHLH1* [14], an R2R3-MYB transcription factor, *CrMYB68* [15], and *CsMADS6* [16]. However, the molecular mechanism for how carotenoid composition extends the variation among cultivars remains unclear.

To advance molecular breeding for the enrichment of carotenoids, especially β-cryptoxanthin, Sugiyama et al. [17] examined quantitative trait locus (QTL) mapping for carotenoid composition using a bi-parental population. QTLs for β-cryptoxanthin and total carotenoid content overlapped at the Gn0005 locus in linkage group 6 of the pollen parent map, and their logarithm (base 10) of odds (LOD) value was around 3.0. This locus was later found to be adjacent to the locus of 9-*cis*-epoxycarotenoid dioxygenase (NCED) by comparing the linkage map and genome sequences of clementine mandarins (https://phytozome.jgi.doe.gov/pz/portal.html) [18] and Satsuma mandarin using the Mikan Genome Database (MiGD, https://mikan.dna.affrc.go.jp/) [19]. Various QTLs for other carotenoid components were also detected, but most of them expressed low LOD values (less than 2.5). Sugiyama et al. [20] extended the

study to compare the genetic locus and expression QTL (eQTL) of carotenoid metabolic genes using a bi-parental population (Fig 1). Against the genetic loci of the major carotenoid metabolic genes, phytoene synthase (*PSY*) on linkage group (LG)-4, phytoene desaturase (*PDS*) on LG-3, ζ-carotene desaturase (*ZDS*) on LG-9, lycopene β-cyclase (*LCYb*), and β-ring hydroxylase (*HYb*) on LG-3, zeaxanthin epoxidase (*ZEP*) on LG-2, and *NCED* on LG-06, along with eQTLs of *PSY*, *HYb*, and *ZEP* could be mapped on the loci of their corresponding genes, revealing that their transcription is regulated primarily by *cis*-elements in their promoter regions. In contrast, the eQTLs of both *PDS* and *ZDS* were mapped around the genetic locus of the Tf0271 DNA marker on LG-8, indicating that transcription would be regulated by

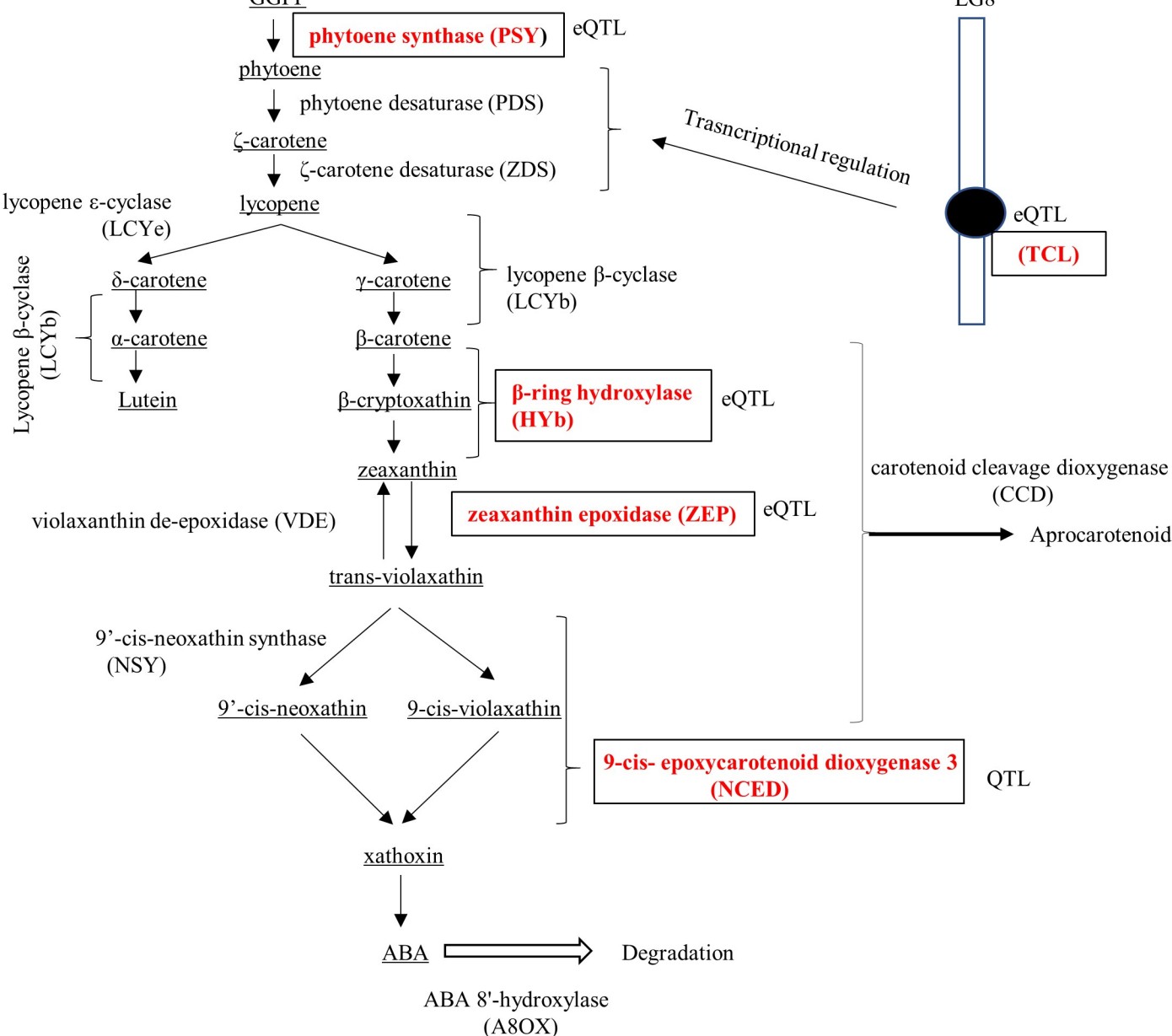

**Fig 1. Carotenoid metabolic pathway in citrus fruits and 5 target genes in allele mining system for the enrichment of β-cryptoxanthin in juice sac tissues of fruits. Notes:** 5 target genes responsible for enrichment of carotenoids in juice sac tissues of fruits are selected from the past QTL and eQTL analyses [15, 17].

*trans*-elements in this genetic region. Thus, the mode of transcriptional regulation revealed a difference in each gene. There have been several reports that allelic differences in carotenoid metabolic genes have been involved in the modulation of carotenogenesis in crops. In wheat grain (*Triticum turgidum* L. subsp. *durum* (Desf.) Husn.), the allelic divergence of *PSY* may be responsible for the grain's yellow pigment content [21]. In maize (*Zea. mays* subsp. *mays* (L.) Iltis), one *PSY* allele with an insertion in its promoter region increased the expression in endosperm and the carotenoid content of yellow maize [22]. In Satsuma mandarins, allelic combination has caused the transcriptional variation of *PSY* and *ZEP* in mature fruit [23, 24]. Considering that transcriptional variation among carotenoid metabolic genes is significantly associated with the carotenoid composition and content among varieties [5] and Japanese cultivars comprise admixture genomes derived from the limited ancestral species [25], it is possible that the allelic combinations among carotenoid metabolic genes would likely be one of the major factors influencing the carotenoid diversity across citrus varieties.

In this study, we applied an allele mining approach to understand the molecular mechanism by which cultivars extend the diversity of carotenoid composition as well as to develop an allele mining system for enrichment of carotenoids, especially β-cryptoxanthin, in juice sac tissues of fruits. Allele mining is a promising technique for dissecting the naturally occurring allelic variation in candidate genes controlling key agronomic traits that have potential applications in crop improvement programs [26]. Among the various carotenoid metabolic genes, the loci of *PSY*, *HYb*, *ZEP*, *NCED*, and a putative causative gene on eQTLs that control the transcription level of *PDS* and *ZDS (TCL)* were selected as major candidates to strongly influence the carotenoid content and composition in juice sac tissues of fruits, as determined from previous studies [15, 17]. The independent alleles of five target genes in 13 ancestral varieties (founder) of the Japanese citrus breeding population were explored based on single nucleotide polymorphisms (SNPs) by SureSelect target enrichment analysis. *PSY*, *HYb*, *ZEP*, and *TCL* possessed 7, 11, 5, and 4 independent alleles in 13 founders, respectively, while some of the 13 founders shared independent alleles derived from ancestral species. A Bayesian statistical analysis was applied to the association analysis between allelic composition and carotenoid composition in 263 breeding pedigrees. Our findings reveal *PSY* and *ZEP* as key genes that influenced the total carotenoid content and β-cryptoxanthin content in juice sac tissues of fruits. We further discussed the ability of the allele mining system to enrich carotenoid content in juice sac tissues of fruits.

## Materials and methods

### Plant materials

All plants used in the experiments were cultivated in the research field of the National Agriculture and Food Research Organization Institute of Fruit and Tea Tree Science, Citrus Research Center, Okitsu, Shizuoka, Japan (Table 1). Genomic DNA was extracted from fresh and fully expanded leaves of these plants according to the method described by Dellaporta *et al.* [27]. Juice sac tissues were collected from mature fruits at their harvest time and were immediately frozen by liquid nitrogen for liquid chromatography analysis of carotenoid components.

### SureSelect target enrichment of carotenoid metabolic genes in 13 founders

Genomic DNA of 13 founders (Table 1) was used for SureSelect target enrichment. The genomic DNA samples were randomly fragmented using a SureSelect $^{QXT}$ Reagent Kit (Agilent Technologies, Santa Clara, CA, USA) and amplified using SureSelect primer Mix (Agilent Technologies). The adapter-attached DNA libraries were hybridized to the SureSelect$^{XT}$ custom library, and the captured DNA was purified using Dynabeads MyOne Streptavidin T1

**Table 1. Plant materials used in this study.**

| Code | Name | Academic name/ Parental combination | Breeding generation |
| --- | --- | --- | --- |
| 1 | Dancy tangerine | *Citrus tangerina hort. ex Tanaka* | Founder |
| 2 | Grapefruit (cv. Duncan) | *C. paradisi* Macfad. | Founder |
| 3 | Kishu mikan (cv. Mukakukishu) | *C. kinokuni* hort. ex Tanaka | Founder |
| 4 | Buntan pumelo (cv. Tanikawa buntan) | *C. grandis* Osbeck | Founder |
| 5 | Hassaku | *C. hassaku* hort. ex Tanaka | Founder |
| 6 | Hyuganatsu | *C. tamurana* hort. ex Tanaka | Founder |
| 7 | Sweet orange (cv. Trovita) | *C. sinensis* (L.) Osbeck | Founder |
| 8 | Iyo | *C. iyo* hort. ex Tanaka | Founder |
| 9 | Kunenbo mandarin | *C. nobilis* Lour. var. kunep Tanaka | Founder |
| 10 | Ponkan mandarin (cv. Yoshida) | *C. reticulata* Blanco | Founder |
| 11 | Willowleaf mandarin | *C. deliciosa* Ten. | Founder |
| 12 | King mandarin | *C. nobilis* Lour. | Founder |
| 13 | Murcott | Hybrid | Founder |
| 14 | Clementine mandarin | *C. clementina* hort. ex Tanaka (11 × 7) | Natural cross between founders (G0) |
| 15 | Satsuma mandarin (cv. Miyagawa-Wase) | *C. unshiu* Marc. (3 × 9) | Natural cross between founders (G0) |
| 16 | Minneola | 2 × 1 | 1st generation (G1) |
| 17 | Seminole | 2 × 1 | 1st generation (G1) |
| 18 | Orland | 2 × 1 | 1st generation (G1) |
| 19 | Southern Yellow | 4 × 3 | 1st generation (G1) |
| 20 | Nankou | 15 × 14 | 1st generation (G1) |
| 21 | Ariake | 7 × 14 | 1st generation (G1) |
| 22 | Sweet Spring | 15 × 5 | 1st generation (G1) |
| 23 | Awa orange | 6 × 7 | 1st generation (G1) |
| 24 | JHG | 15 × 6 | 1st generation (G1) |
| 25 | Kiyomi | 15 × 7 | 1st generation (G1) |
| 26 | Aki-tangor | 15 × 7 | 1st generation (G1) |
| 27 | HF9 | 15 × 7 | 1st generation (G1) |
| 28 | Hayaka | 15 × 10 | 1st generation (G1) |
| 29 | Kankitsu Chukanbohon Nou 6 Gou | 12 × 3 | 1st generation (G1) |
| 30 | Kara | 15 × 12 | 1st generation (G1) |
| 31 | Encore | 12 × 11 | 1st generation (G1) |
| 32 | Wilking | 12 × 11 | 1st generation (G1) |
| 33 | Page | 16 × 14 | 2nd generation (G2) |
| 34 | Robinson | 14 × 18 | 2nd generation (G2) |
| 35 | Lee | 14 × 18 | 2nd generation (G2) |
| 36 | Fairchild | 14 × 18 | 2nd generation (G2) |
| 37 | Fortune | 14 × 18 | 2nd generation (G2) |
| 38 | Nova | 14 × 18 | 2nd generation (G2) |
| 39 | Osceola | 14 × 18 | 2nd generation (G2) |
| 40 | Seihou | 25 × 16 | 2nd generation (G2) |
| 41 | Akemi | 25 × 17 | 2nd generation (G2) |
| 42 | Okitsu 46 gou | 22 × 7 | 2nd generation (G2) |
| 43 | Nishinokaori | 25 × 7 | 2nd generation (G2) |
| 44 | KyOw No.21 | 25 × 15 | 2nd generation (G2) |
| 45 | KyOw No.14 | 25 × 15 | 2nd generation (G2) |
| 46 | Tsunokaori | 25 ×15 | 2nd generation (G2) |
| 47 | Shiranuhi | 25 × 10 | 2nd generation (G2) |

(*Continued*)

**Table 1.** (Continued)

| Code | Name | Academic name/ Parental combination | Breeding generation |
|---|---|---|---|
| 48 | Youkou | $25 \times 10$ | 2nd generation (G2) |
| 49 | Harumi | $25 \times 10$ | 2nd generation (G2) |
| 50 | Setomi | $25 \times 10$ | 2nd generation (G2) |
| 51 | EnOw No.21 | $31 \times 15$ | 2nd generation (G2) |
| 52 | Kuchinotsu 39 gou | $31 \times 15$ | 2nd generation (G2) |
| 53 | Miho-core | $15 \times 31$ | 2nd generation (G2) |
| 54 | KyEn No.4 | $25 \times 31$ | 2nd generation (G2) |
| 55 | KyEn No.5 | $25 \times 31$ | 2nd generation (G2) |
| 56 | Tsunonozomi | $25 \times 31$ | 2nd generation (G2) |
| 57 | Amaka | $25 \times 31$ | 2nd generation (G2) |
| 58 | Okitsu 45 gou | $25 \times 32$ | 2nd generation (G2) |
| 59 | Tamami | $25 \times 32$ | 2nd generation (G2) |
| 60 | Benibae | $27 \times 31$ | 2nd generation (G2) |
| 61 | Hareyaka | $31 \times 10$ | 2nd generation (G2) |
| 62 | Amakusa | $45 \times 33$ | 3rd generation (G3) |
| 63 | Kuchinotsu 38 gou | $44 \times 34$ | 3rd generation (G3) |
| 64 | Kankitsu Chukanbohon Nou 5 Gou | $35 \times 3$ | 3rd generation (G3) |
| 65 | E-647 | $25 \times 39$ | 3rd generation (G3) |
| 66 | Southern Red | $30 \times 51$ | 3rd generation (G3) |
| 67 | Kuchinotsu 28 gou | $44 \times 1$ | 3rd generation (G3) |
| 68 | Haruhi | $42 \times 23$ | 3rd generation (G3) |
| 69 | 2700OIy-25 | $43 \times 8$ | 3rd generation (G3) |
| 70 | No.1408 | $51 \times (25 \times 8)$ | 3rd generation (G3) |
| 71 | Kuchinotsu 18 gou | $44 \times 31$ | 3rd generation (G3) |
| 72 | Kuchinotsu 35 gou | $44 \times 31$ | 3rd generation (G3) |
| 73 | Kanpei | $43 \times 10$ | 3rd generation (G3) |
| 74 | Okitsu 57 gou | $42 \times 49$ | 3rd generation (G3) |
| 75 | Asumi | $42 \times 49$ | 3rd generation (G3) |
| 76 | Asuki | $42 \times 49$ | 3rd generation (G3) |
| 77 | Seinannohikari | $51 \times 48$ | 3rd generation (G3) |
| 78 | Kuchinotsu 27 gou | $51 \times 48$ | 3rd generation (G3) |
| 79 | Kuchinotsu 33 gou | $45 \times 31$ | 3rd generation (G3) |
| 80 | Tsunokagayaki | $45 \times 31$ | 3rd generation (G3) |
| 81 | Setoka | $54 \times 13$ | 3rd generation (G3) |
| 82 | Kuchinotsu 36 gou | ns $\times 13$ | 3rd generation (G3) |
| 83 | Reikou | ns $\times 13$ | 3rd generation (G3) |
| 84 | Ehime Kashi No.28 | $20 \times 62$ | 4th generation (G4) |
| 85 | Kuchinotsu 49 gou | $63 \times 34$ | 4th generation (G4) |
| 86 | Okitsu 56 gou | $58 \times 64$ | 4th generation (G4) |
| 87 | No.1011 | $20 \times 69$ | 4th generation (G4) |
| 88 | Kuchinotsu 51 gou | ns $\times 78$ | 4th generation (G4) |
| 89 | Harehime | $65 \times 15$ | 4th generation (G4) |
| 90 | Okitsu 63 gou | $65 \times 10$ | 4th generation (G4) |
| 91 | Kuchinotsu 52 gou | $80 \times 21$ | 4th generation (G4) |
| 92 | Mihaya | $56 \times 70$ | 4th generation (G4) |
| 93 | Kuchinotsu 54 gou | $82 \times 29$ | 4th generation (G4) |
| 94 | 080716 | $84 \times 77$ | 5th genaration (G5) |

**Table 1.** (Continued)

| Code | Name | Academic name/ Parental combination | Breeding generation |
|------|------|-------------------------------------|---------------------|
| 95 | Okitsu 67 gou | 89 × 86 | 5th genaration (G5) |
| ~105 | 10 strains | 29 × 7 | Hybrid Seedling |
| ~120 | 15 strains | 26 × 19 | Hybrid Seedling |
| ~138 | 18 strains | 11 × 88 | Hybrid Seedling |
| ~154 | 16 strains | *C. tankan* Hayata × 71 | Hybrid Seedling |
| ~175 | 21 strains | 95 × 71 | Hybrid Seedling |
| ~201 | 26 strains | 81 × 70 | Hybrid Seedling |
| ~212 | 11 strains | 68 × Soren tangero | Hybrid Seedling |
| ~232 | 20 strains | 7 × 65 | Hybrid Seedling |
| ~244 | 12 strains | 68 × 3 | Hybrid Seedling |
| ~266 | 22 strains | 26 × 52 | Hybrid Seedling |
| ~269 | 3 strains | 76 × 11 | Hybrid Seedling |
| ~275 | 6 strains | 14 × 4 | Hybrid Seedling |

beads (Thermo Fisher, Waltham, MA, USA). The nucleic acids of the target carotenoid metabolic genes (*PSY*, *HYb*, *ZEP*, and *NCED*) were enriched with index primers. Index primers for each target gene were designed using the clementine mandarin genome sequence ver. 1.0 as reference (https://phytozome.jgi.doe.gov/pz/portal.html), as follows: *PSY* (Ciclev10011841m. g), *HYb* (Ciclev10005481m.g), *ZEP* (Ciclev10025089m.g), and *NCED* (Ciclev10019364m.g). The enriched fragments were purified using Agencourt AMPure XP (Beckman Coulter, Brea, CA, USA). The purified products were subjected to next-generation sequencing (NGS) analysis in a single lane on a HiSeq2500 system (Illumina, San Diego, CA, USA) with a paired-end read length of 100 bp. Low-quality bases and Illumina sequencing adapters were trimmed using cutadapt v1.1 (https://cutadapt.readthedocs.org/en/stable/) and Trimmomatic v0.32 (http://www.usadellab.org/cms/index.php?page=trimmomatic). Trimmed reads were mapped to the clementine genome sequence v1.0 using BWA v0.7.10 (http://bio-bwa.sourceforge.net/), GATK Lite v2.3.0 (https://www.broadinstitute.org/gatk/), and Picard v1.133 (http://broadinstitute.github.io/picard/). Mutation analysis was carried out using the Samtools v1.2 (http://www.htslib.org/man/Samtools/) and BCFtools ver. 1.2 (http://www.htslib.org/man/bcftools/). Library preparation, Illumina PE sequencing, mapping, and mutation analyses of PE reads were performed by Hokkaido System Science Co., Ltd. (Sapporo, Hokkaido, Japan).

## SNP genotyping by GoldenGate assay and Fluidigm assay

SNP genotyping was carried out using the GoldenGate assay (Illumina) and a Fluidigm assay (Fluidigm, South San Francisco, CA, USA). For the GoldenGate assay, bead arrays were designed using the Illumina® Assay Design Tool according to the manufacturer's instructions. SNP genotyping analysis was performed using the Goldengate Assay system on an iScan system (Illumina). The assay was performed according to the manufacturer's instructions. The scanned image data were converted to genotype scores using the Genome Studio software by a function of the Genotyping module (Illumina).

For the Fluidigm assay, TaqMan minor groove binder (TaqMan-MGB) probe and primer sets were designed using Primer Express ver. 3.0 (Applied Biosystems, Foster City, CA, USA). 5-carboxyl-fluorescein (FAM) and hexachloro-fluorescein (HEX) were used to label the 5-end of the oligonucleotides. Genotyping was carried out using SNPType chemistry (Fluidigm Corp., South San Francisco, CA, USA) on a Fluidigm Nanofluidic 96.96 Dynamic Array [28],

according to the manufacturer's instructions. Thermal cycling comprised an initial thermal mix cycle (70˚C for 30 min; 25˚C for 10 min), followed by a hot-start *Taq* polymerase activation step (94˚C for 15 min) and a touchdown amplification protocol, as follows: 10 cycles of 94˚C for 20 s, 65˚C for 1 min (decreasing 0.8˚C per cycle); then 26–46 cycles of 94˚C for 20 s, 57˚C for 1 min, with a hold at 20˚C for 30 s after every four cycles to collect end-point fluorescent images of the chip using the Biomark imager (Fluidigm Corp). Data were analyzed using the Fluidigm SNP Genotyping Analysis Software with dye settings MGB-FAM and MGB-HEX.

### Filtering of inheritable SNPs in parent-offspring trios

The SNPs detected by the GoldenGate assay and Fluidigm assay were used to evaluate the inheritability and repulsion in gametes using MARCO software [29] at the condition without any discrepancy among 78 combinations of parent-offspring trios. A reliable SNP with confirmed heritability between parent and offspring was designated as a trio-tagged SNP.

### Quantification of the carotenoid content in breeding materials

Quantification of eight representative carotenoids [phytoene, α-carotene, ζ-carotene, lutein, β-carotene, β-cryptoxanthin, zeaxanthin, and violaxanthin] was carried out according to a modification of the method proposed by Kato *et al.* [5]. Juice sac tissues were homogenized and extracted in a solution of hexane, acetone, and ethanol (50:25:25, v/v). The pigments partitioned into the hexane phase evaporated until dry. They were then dissolved in methyl tert-butyl ether containing 0.1% (w/v) 2,6-di-tertbutyl-4-methylphenol. The extracts containing carotenoids esterified to fatty acids were saponified with 10% (w/v) methanolic KOH. After saponification, water-soluble extracts were removed from the extract by adding NaCl-saturated water. The pigments were repartitioned into 2mL of the MTBE phase. An aliquot (20 μL) was obtained using a reverse-phase HPLC system (Jasco, Tokyo, Japan) fitted with a YMC Carotenoid S-5 column of $250 \times 4.6$-mm mm i.d. (Waters, Milford, MA) at a flow rate of 1 mL min$^{-1}$. The eluent was monitored using a photodiode array detector (MD-910, Jasco, Tochigi, Japan). Each sample was analyzed using the gradient elution schedule proposed by Rouseff et al. [30]. The composition was alternated from 90% methanol, 5% MTBE, and 5% water through a linear gradient to 95% MeOH and 5% MTBE over 12 min, then to 86% MeOH and 14% MTBE over 8 min, 75% MeOH, and 25% MTBE over 10 min, and 50% MeOH and 50% MTBE over 20 min. The peaks were identified by comparing their specific retention times and absorption spectra with authentic standards. The concentrations of the standard solutions were estimated on the basis of the absorption coefficient at 286 nm for phytoene, 400 nm for ζ-carotene, 452 nm for t-violaxanthin, c-violaxanthin, lutein, β-cryptoxanthin, α-carotene, and zeaxanthin, and 453 nm for β-carotene [31]. The sample concentrations were estimated from standard curves. Violaxanthin and ζ-carotene were obtained as the sum of isomers and total CARs as the sum of the carotenoids. The carotenoid concentration was estimated by the standard curves and expressed as milligrams per hundred fresh weight grams (mg/100FWG).

### Estimation for the genetic effects of alleles in the target gene regions by using a Bayesian regression model

To evaluate the genetic effects of alleles in the target genetic loci on carotenoid contents, we applied a multi-loci Bayesian linear regression model [34]. The observed phenotypic value for

the $i^{th}$ variety $y_i$ is described as follows:

$$y_i = \alpha + \sum_{j}^{J} \sum_{l}^{L_j} \gamma_j (x_{ijl} + x'_{ijl}) \beta_{jl} + e_i, \qquad (Eq\ 1)$$

where $\alpha$ represents the intercept. $L_j$ represents the number of alleles in the target gene region $j$ ($j = 1,2,\ldots,J$). $x_{ijl}(x'_{ijl})$ denotes the maternal (paternal) allele in the target gene region $j$ for variety $i$ and equals to 1 if the maternal (paternal) allele is the $l^{th}$ allele ($l = 1,\ldots,L_j$) and 0 otherwise. $\gamma_j$ signifies the indicator variable and $\gamma_j = 1$ corresponds to the case where gene region $j$ is included in the model as a QTL representative and $\gamma_j = 0$ implies exclusion. $\beta_{jl}$ denotes the allele effect associated with the allele $l$ for gene region $j$, which was assumed to follow $N(0, \sigma^2_{\beta_j})$. The residual error $e_i$ was assumed to follow $N(0, \sigma^2_e)$. The genetic variance of gene region $j$, i.e., $\sigma^2_{\beta_j}$, and the residual variance, i.e., $\sigma^2_e$, were assumed to follow the scaled inverse chi-square distributions Inv–$\chi^2(v_\beta, S_\beta)$ and Inv–$\chi^2(v_e, S_e)$, respectively, as described by Iwata et al. [32]. A prior probability for $\gamma_j$ was assumed to follow the Bernoulli distribution with probability $\pi = \frac{\lambda}{J}$, as described by Iwata et al. [32]. Estimation of the parameters in the above model via the Bayesian regression was conducted as described by Iwata et al. [32]. MCMC sampling was used for the Bayesian inference on each parameter. For each dataset, MCMC cycles were repeated $13 \times 10^4$ times. The first $3 \times 10^4$ (burn-in) cycles were not used for the parameter estimation. The sampling was conducted every ten cycles to reduce serial correlation, such that the total number of samples we retained was $1 \times 10^4$. The hyperparameters of the model were set as $v_\beta = 4$, $S^2_\beta = 0.04$, $v_e = -2$, $S^2_e = 0$ and $\pi = 0.6$ ($\lambda = 3$) to correspond to the model known as BayesB. This sampling scheme was based on a previously described evaluation of the convergence of MCMC cycles [32, 33]. In this analysis, we evaluated the allele effects ($\zeta_{jl}$) in the target gene region by multiplying the product of the posterior means of $\gamma_j$ and $\beta_{jl}$. Further, we calculated the proportion of the expected variance explained by all alleles, the allele which had the highest effect, and the allele which had the largest variance to the phenotypic variance as the allele contribution. Significant gene regions were determined by the permutation test as described by Iwata et al. [34].

### Minimal set of trio-tagged SNPs to discriminate the independent alleles of *PSY* and *ZEP* in 13 founders of Japanese breeding pedigrees

Minimal set of trio-tagged SNPs was calculated to discriminate the independent alleles of *PSY* and *ZEP* in 13 founders using MinimalMarker software [35].

## Results

### SureSelect target enrichment of carotenoid metabolic genes among the 13 founders

In previous studies of QTL and eQTL analyses using bi-parental populations [17, 20], each genetic locus for *PSY*, *HYb*, *ZEP*, and *NCED*, and eQTL for *PDS* and *ZDS* in linkage group 8 (*TCL*) were shown to have statically effects on carotenoid content and transcription level of carotenoid metabolic genes. In this study, we focused on them and explored an optimum allele associated with high carotenoid content in juice sac tissues of fruits, from a total of 26 alleles present in 13 ancestral species of the Japanese breeding population. Citrus varieties comprised complex mosaic genome structures of a limited number of ancient species via repeated natural cross [36]; therefore, the number of independent alleles that were shared by some of the 13 founders was estimated for five target genes based on SNPs. Since the genomic sequences of

most founders are not available in the public database, a SureSelect target enrichment system was applied to acquire the SNP information of the target genes except for *TCL*. The SNP information was collected from the gene region because sequence variation in promoter region were complicate among 13 founders by the combination of insertion, deletion and point mutation among citrus varieties and it would be unfavorable to extract stably inherited SNPs among citrus varieties and their progenies. Because carotenoid metabolic genes have several isoforms in the genome, genome sequences in the gene region of Ciclev10011841m.g for *PSY*, Ciclev10005481m.g for *HYb*, Ciclev10025089m.g for *ZEP*, and Ciclev10014639m.g for *NCED* in the clementine mandarin genome sequence ver 1.0 were utilized to create custom bait libraries that covered the genomic region around target genes. These loci were confirmed to influence the carotenoid content and transcription levels of metabolic genes in previous studies. The constructed captured libraries enriching four metabolic genes were sequenced using a HiSeq2500 system (Illumina), and NGS data were aligned with the reference sequence of clementine mandarin genome sequence ver 1.0 using CLC Genomic Workbench 6.5.1 (CLC bio, Aarhus, Denmark). Numerous SNPs and indels were found in 4 target genes when overviewing 13 founders. The number of SNPs in *PSY*, *HYb*, *ZEP*, and *NCED* with more than readable SNP quality score (>150) according to manufacturer's description were 107, 31, 54 and 19 respectively, when NGS data of 13 founders were compared with the corresponding reference gene of clementine mandarin. For *TCL*, the causative gene that controls the transcription levels of *PDS* and *ZDS* has not been characterized. Therefore, arbitrary SNPs on *TCL* were applied to estimate the number of independent alleles for uncharacterized causative genes on *TCL* given that the allele on common haplotype blocks derived from common founders would reveal the same SNP genotypes. The candidate SNPs on *TCL* were explored using the TASUKE program in MiGD [19]. Based on the SNP information, 17 SNP markers for *PSY*, 15 for *HYb*, 31 for *ZEP*, 8 *NCED*, and 5 for *TCL* were developed for a SNP genotyping assay using the Golden-Gate assay system (Illumina) and Fluidigm BioMark™ HD assay system (Fluidigm). PSY catalyzes the first committed condensation step from GGPP to produce the first C40 carotene, phytoene. Carotenoid composition is highly influenced by the transcription balance among carotenoid metabolic genes [5]. In previous QTL and eQTL analyses, *PSY*, *HYb*, *ZEP*, *NCED*, and eQTL for *PDS* and *ZEP* have been shown to increase carotenoid content in juice sac tissues of fruits [17, 20]. These genes including a causative gene on *TCL* (denoted in red font within boxes) are the targets for allele mining in this study. PSY, phytoene synthase; PDS, phytoene desaturase; ZDS, ζ-carotene desaturase; LCYe, lycopene ε-cyclase; LCYb, lycopene β-cyclase; HYb, β-carotene hydroxylase; ZEP, zeaxanthin epoxidase; VDE, violaxanthin de-epoxidase; NSY, neoxanthin synthase; CCD, carotenoid cleavage dioxygenase; NCED, 9-*cis*-epoxycarotenoid dioxygenase.

## Independent alleles in 13 founders based on the SNP genotype information of reliable SNP markers

A total of 76 SNP markers were applied to the genotyping of 13 founders and 78 offspring that had parent–offspring relationships among them. Modern citrus varieties reveal extensive sharing of haplotypes from ancient species [37]. Considering the admixture genomes and long cultivation history of 13 founders, numerous *de novo* mutations frequently occur in those sharing haplotypes; these would hamper the identification of a synonymous allele derived from common ancient species, out of the 26 alleles present in the 13 founders. Trio analysis was carried out to remove SNPs of *de novo* mutations and obtain reliable SNPs on the basis of inheritance in the lineages. Seventy-eight combinations of parent-offspring varieties were evaluated by the computer software "MARCO". In total, 57 SNPs passed through the filtering by trio analysis

and were valid as trio-tagged SNPs. Schematic diagram to generate 57 trio-tagged SNPs and their detail information are summarized in S1 Fig and S1 Table, respectively. Tentatively, each of the 26 alleles calculated in the13 founders was also named according to the style of abbreviation combination for gene, variety, and allele number (1 or 2), i.e., *PSY* alleles in grapefruit were named *PSY-a-gf1 and PSY-a-gf2*, and so on. The alleles with identical trio-tagged SNP genotypes were assigned as independent alleles, and 12, 10, 24, and 6 trio-tagged SNPs were used to find an independent allele for *PSY*, *HYb*, *ZEP*, and *NCED* alleles, respectively. The numbers of independent alleles in 13 founders were as follows: 7 alleles for *PSY*, 7 alleles for *HYb*, 11 alleles for *ZEP*, 5 alleles for *NCED*, and 4 alleles *for TCL*. The members of each synonymous allele and their genotypes of trio-tagged SNPs are summarized in Table 2. For example, the *PSY*-independent allele was defined based on the genotype of 12 trio-tagged SNPs, and there were 7 independent alleles from *PSY-a* to *PSY-g*. *PSY-a-gf2* in grapefruit, *PSY-a-ks1* in Kishu mikan, *PSY-a-hs2* in hassaku, *PSY-a-so2* in sweet orange, *PSY-a-kb1* in Kunenbo mandarin, *PSY-a-wl2* in willow leaf mandarin, *PSY-a-kg2* in King mandarin, and *PSY-a-mch* in 'Murcott' revealed the same SNP genotype and were assigned to *PSY-a*. The trio-tagged SNP genotypes for *PSY*, *HYb*, *ZEP*, *NCED*, and *TCL* are summarized in S2 Table.

## Genotyping of trio-tagged SNPs in breeding populations and the frequency of alleles in the progress of breeding generation

SNP genotyping assay was carried out for 275 plant materials, including13 founders, and their allelic composition was characterized based on the SNP genotype data of 57 trio-tagged SNPs. The allelic genotypes of five target genes in major cultivars are summarized in S3 Table. There was no discrepancy in the transmission of all independent alleles from parents to offspring among the samples in the 78 parent–offspring combinations. Breeding populations have been generated by crossing with one another, and the most advanced breeding generations of the registered cultivars belong to the 4th generation from 13 founders in the breeding program. The numbers of the registered cultivars and preserved selection lines in each generation were as follows: founder, 13 varieties; zero generation (G0): 2 (Satsuma mandarin and Clementine mandarin), 1st generation (G1), 14 varieties and lines (i.e. 'Kiyomi', 'Sweet spring', etc.); 2nd generation (G2), 23 (i.e. 'Harumi', 'Tamami', etc.); 3rd generation (G3), 21 (i.e. 'Setoka', 'Tsunokagayaki', etc.); 4th generation (G4), 10 (i.e. 'Harehime', 'Ehime Kashi No. 28', etc.). In Japanese breeding programs, the enrichment of carotenoids in fruit is one of the most important breeding traits. The allele frequencies of the five target genes were traced during those breeding generations, except for G0 because Satsuma mandarin and clementine mandarin were generated by the natural cross of founders between Kunenbo mandarin and Kishu mikan or between Willowleaf mandarin and King mandarin, respectively (Fujii et al. 2016). Interestingly, several alleles of the five target genes were selected out in progress of breeding generation. For example, out of seven *PSY* alleles in founders, *PSY-g* and *PSY-f* disappeared from G2, and the occupied ratio of *PSY-c* decreased from 15.4% in founders to 5% in G4 (Fig 2A). Of the seven *HYb* alleles in founders, *HYb-c*, *HYb-e*, and *HYb-f* disappeared in the G4, and the occupied ratio of *HYb-g* increased from 3.8% in founders to 27.8% in G4 (Fig 2B). Of the 11 *ZEP* alleles in founders, *ZEP-f*, *ZEP-g*, *ZEP-h*, *ZEP-j*, and *ZEP-k* disappeared in G4, and the occupied ratio of *ZEP-e* increased from 7.7% in founders to 26.3% in G4 (Fig 2C). Of the five *NCED* alleles in founders, *NCED-d* and *NCED-e* disappeared in G4, and the occupied ratio of *NCED-c* reduced from 15.4% in founders to 5.0% in G4 (Fig 2D). Additionally, out of 4 *TCL* alleles in founders, *TCL-d* disappeared in G4, and the occupied ratio of *TCL-a* decreased from 38.5% in founders to 12.1% in G4 (Fig 2E). Thus, several alleles in founders disappeared during

**Table 2. Allelic composition of independent alleles of 5 target genes including a causative gene on TCL in 13 founders.**

| PSY Independent allele | PSY Founder | PSY Tentative allele | HYb Independent allele | HYb Founder | HYb Tentative allele | ZEP Independent allele | ZEP Founder | ZEP Tentative allele | NCED Independent allele | NCED Founder | NCED Tentative allele | TCL Independent allele | TCL Founder | TCL Tentative allele |
|---|---|---|---|---|---|---|---|---|---|---|---|---|---|---|
| PSY-a | Grapefruit | PSY-a-gf2 | HYb-a | Dancy tangerine | HYb-a-dc1 | ZEP-a | Dancy tangerine | ZEP-a-dc1 | NCED-a | Dancy tangerine | NCED-a-dc1 | TCL-a | Buntan pumelo | TCL-a-tbh |
| | Kishu mikan | PSY-a-ks1 | | Kishu mikan | HYb-a-ks2 | | Kishu mikan | ZEP-a-ks1 | | Grapefruit | NCED-a-gf2 | | Hassaku | TCL-a-hsh |
| | Hassaku | PSY-a-hs2 | | Buntan pumelo | HYb-a-tb2 | | Hassaku | ZEP-a-hs1 | | Buntan pumelo | NCED-a-tb2 | | Hyuganatsu | TCL-a-hg2 |
| | Sweet orange | PSY-a-so2 | | Hassaku | HYb-a-hs2 | | Hyuganatsu | ZEP-a-hg1 | | Hassaku | NCED-a-hsh | | Sweet orange | TCL-a-so2 |
| | Kunenbo mandarin | PSY-a-kb1 | | Iyo | HYb-a-iy2 | | Sweet orange | ZEP-a-so1 | | Hyuganatsu | NCED-a-hg1 | | Iyo | TCL-a-iy2 |
| | Willowleaf mandarin | PSY-a-wl2 | | Kunenbo mandarin | HYb-a-kb2 | | Iyo | ZEP-a-iy1 | | Sweet orange | NCED-a-so2 | | Kunenbo | TCL-a-kb2 |
| | King mandarin | PSY-a-kg2 | | Ponkan mandarin | HYb-a-pk1 | | Kunenbo mandarin | ZEP-a-kb1 | | Iyo | NCED-a-iy1 | | King mandarin | TCL-a-kg2 |
| | Murcott | PSY-a-mch | | Murcott | HYb-a-mc2 | | Ponkan mandarin | ZEP-a-pkh | | Kunenbo mandarin | NCED-a-kb1 | | Murcott | TCL-a-mc2 |
| PSY-b | Iyo | PSY-b-iy1 | HYb-b | Grapefruit | HYb-b-gf2 | | Willowleaf mandarin | ZEP-a-wlh | | Ponkan mandarin | NCED-a-pk1 | TCL-b | Dancy tangerine | TCL-b-dc1 |
| | Ponkan mandarin | PSY-b-pk1 | | Buntan pumelo | HYb-b-tb1 | ZEP-b | Dancy tangerine | ZEP-b-dc2 | | King mandarin | NCED-a-kg1 | | Grapefruit | TCL-b-gf1 |
| | Willowleaf mandarin | PSY-b-wl1 | | Hyuganatsu | HYb-b-hgh | | Kishu mikan | ZEP-b-ks2 | | Murcott | NCED-a-mch | | Iyo | TCL-b-iy1 |
| | King mandarin | PSY-b-kg1 | | Sweet orange | HYb-b-so1 | | Murcott | ZEP-b-mc2 | NCED-b | Hyuganatsu | NCED-b-hg2 | | Murcott | TCL-b-mc1 |
| | Dancy tangerine | PSY-b-dc1 | | Kunenbo mandarin | HYb-b-kb1 | ZEP-c | Kunenbo mandarin | ZEP-c-kb2 | | Kunenbo mandarin | NCED-b-kb2 | | Sweet orange | TCL-b-so1 |
| | Buntan pumelo | PSY-b-tb2 | | Willowleaf mandarin | HYb-b-wlh | | King mandarin | ZEP-c-kg2 | | Kishu mikan | NCED-b-ksh | | Ponkan mandarin | TCL-b-pk1 |
| PSY-c | Grapefruit | PSY-c-gf1 | | King mandarin | HYb-b-kg1 | ZEP-d | Buntan pumelo | ZEP-d-tb2 | | Willowleaf mandarin | NCED-b-wlh | | Willowleaf mandarin | TCL-b-wl1 |
| | Buntan pumelo | PSY-c-tb1 | HYb-c | Sweet orange | HYb-c-so2 | | Iyo | ZEP-d-iy2 | NCED-c | Dancy tangerine | NCED-c-dc2 | TCL-C | Kishu mikan | TCL-c-ksh |
| | Hyuganatsu | PSY-c-hg1 | | Murcott | HYb-c-mc1 | ZEP-e | King mandarin | ZEP-e-kg1 | | Iyo | NCED-c-iy2 | | Dancy tangerine | TCL-c-dc2 |
| | Sweet orange | PSY-c-so1 | | King mandarin | HYb-c-kg2 | | Murcott | ZEP-e-mc1 | | Ponkan mandarin | NCED-c-pk2 | | Hyuganatsu | TCL-c-hg1 |
| PSY-d | Kishu mikan | PSY-d-ks2 | HYb-d | Dancy tangerine | HYb-d-dc2 | ZEP-f | Grapefruit | ZEP-f-gf1 | | King mandarin | NCED-c-kg2 | | Kunenbo | TCL-c-kb1 |
| | Iyo | PSY-d-iy2 | | Ponkan mandarin | HYb-d-pk2 | ZEP-g | Grapefruit | ZEP-g-gf2 | NCED-d | Grapefruit | NCED-d-gf1 | | Ponkan | TCL-c-pk2 |
| | Kunenbo mandarin | PSY-d-kb2 | HYb-e | Iyo | HYb-e-iy1 | ZEP-h | Hassaku | ZEP-h-hs2 | | Sweet orange | NCED-d-so1 | | Willowleaf mandarin | TCL-c-wl2 |
| PSY-e | Dancy tangerine | PSY-e-dc2 | | Hassaku | HYb-e-hs1 | ZEP-i | Sweet orange | ZEP-i-so2 | NCED-e | Buntan pumelo | NCED-e-tb1 | | King mandarin | TCL-c-kg1 |
| | Ponkan mandarin | PSY-e-pk2 | HYb-f | Grapefruit | HYb-f-gf1 | ZEP-j | Buntan pumelo | ZEP-j-tb1 | | | | TCL-d | Grapefruit | TCL-d-gf2 |
| PSY-f | Hassaku | PSY-f-hs1 | HYb-g | Kishu mikan | HYb-g-ks1 | ZEP-k | Hyuganatsu | ZEP-k-hg2 | | | | | | |
| PSY-g | Hyuganatsu | PSY-g-hg2 | | | | | | | | | | | | |

**Notes:** *PSY*, phytoene synthase gene; *HYb*, β-ring hydroxylase gene; *ZEP*, zeaxanthin epoxidase gene; *NCED*, 9-*cis*-epoxycarotenoid dioxygenase gene; *TCL*, a putative causative gene on eQTLs that control the transcription level of phytoene desaturase (*PDS*) and ζ-carotene desaturase (*ZDS*). The numbers of independent alleles in 13 founders were as follows: 7 alleles for *PSY*, 7 alleles for *HYb*, 11 alleles for *ZEP*, 5 alleles for *NCED*, and 4 alleles for *TCL*. The members of each synonymous allele are summarized. For example, 7 PSY-independent alleles were named from *PSY-a* to *PSY-g*. Each of the 26 alleles in the13 founders was also named according to the style of abbreviation combination for allele, variety, and allele number (1 or 2), i.e., PSY alleles in grapefruit were named *PSY-c-gf1* and *PSY-a-gf2*, and so on. *PSY-a-gf2* in grapefruit, *PSY-a-ks1* in Kishu mikan, *PSY-a-hs2* in hassaku, *PSY-a-so2* in sweet orange, *PSY-a-kb1* in Kunenbo mandarin, *PSY-a-wl2* in willow leaf mandarin, *PSY-a-kg2* in King mandarin, and *PSY-a-mch* in 'Murcott' revealed the same SNP genotype and were assigned to *PSY-a*. The trio-tagged SNP genotypes for *PSY*, *HYb*, *ZEP*, *NCED*, and *TCL* are summarized in S2 Table.

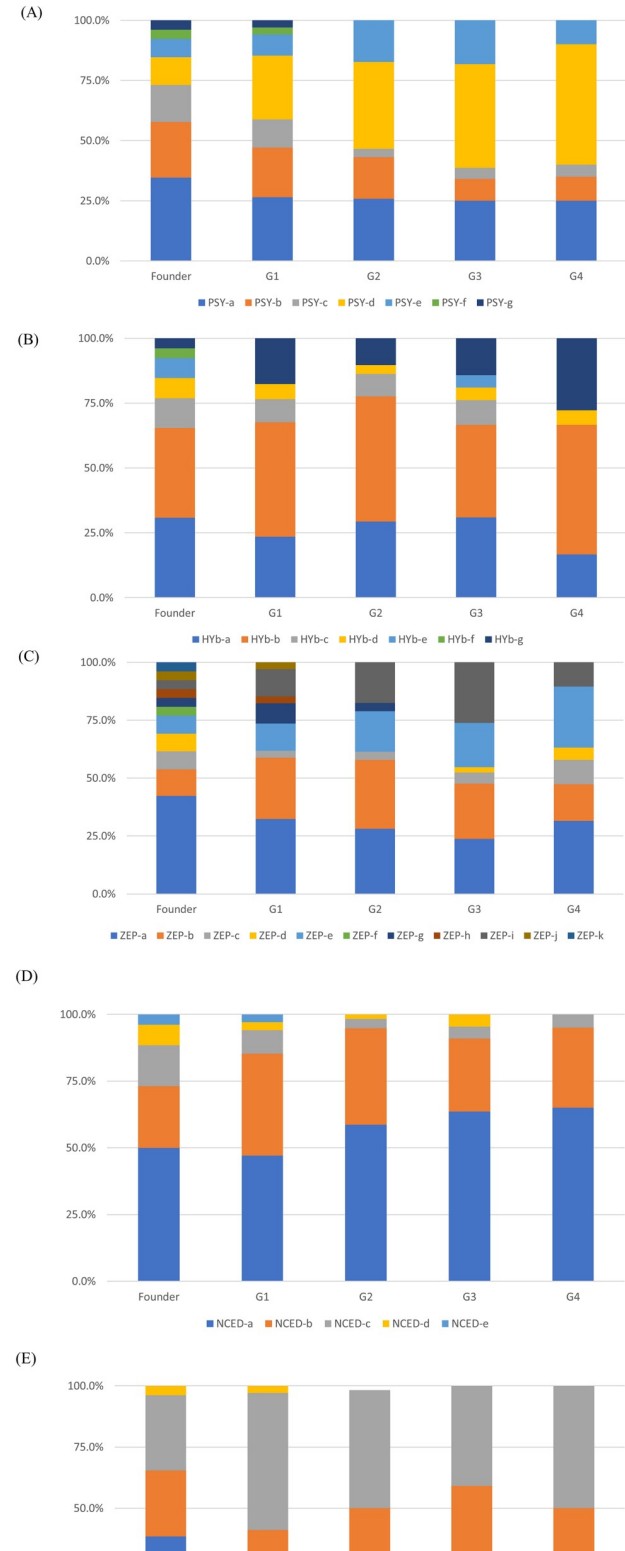

**Fig 2. The frequency of independent alleles in 13 founders during the progress of Japanese breeding population. Notes:** The Japanese breeding population was developed from the germplasms of 13 founders. Several alleles of *PSY* (A), *HYb* (B), *ZEP* (C), *NCED* (D), and *TCL* (E) disappeared in the progressive breeding generation. G1: 1st generation, G2: 2nd generation, G3: 3rd generation, G4: 4th generation. A several alleles disappear in the progress of breeding generation in each gene.

the breeding generation although these disappeared alleles may be revived by crossing with old varieties.

## Quantification of carotenoid components in juice sac tissues of mature fruits from 263 breeding pedigrees

Phytoene, t-violaxanthin, c-violaxanthin, lutein, β-cryptoxanthin, α-carotene, zeaxanthin, and β-carotene in the juice sac tissues of the mature fruits were measured among 263 breeding pedigrees including 11 founders, for which fruits were available, by high-pressure liquid chromatography (HPLC). The content of total carotenoid, β-cryptoxanthin and violaxanthin in the major cultivars is summarized in S3 Table. Fig 3 shows the carotenoid composition for each breeding generation from G1 to G4. The mean total carotenoid content in the 13 founders was 2.2 mg/100FWG, with a range 0.0 mg/100FWG to 6.0 mg/100FWG. The variance in total carotenoid content was 3.9. The fruits of Kishu mikan, King mandarin and 'Murcott' accumulated high concentrations of carotenoids and those of grapefruit, Hassaku and Hyuganatsu accumulated lower concentrations of carotenoids. The major carotenoids present in the 13 founders were either β-cryptoxanthin or violaxanthin, and they comprised more than 67.0% of the total carotenoids. In the breeding population, the mean total carotenoid content was 2.8 mg/100FWG, with a range of 0.2 mg/100FWG to 9.6 mg/100FWG. The variance of total carotenoid content was 2.7, which was less than that of the 13 founders. These results indicated that the variance of total carotenoids became smaller in the breeding population, whereas total

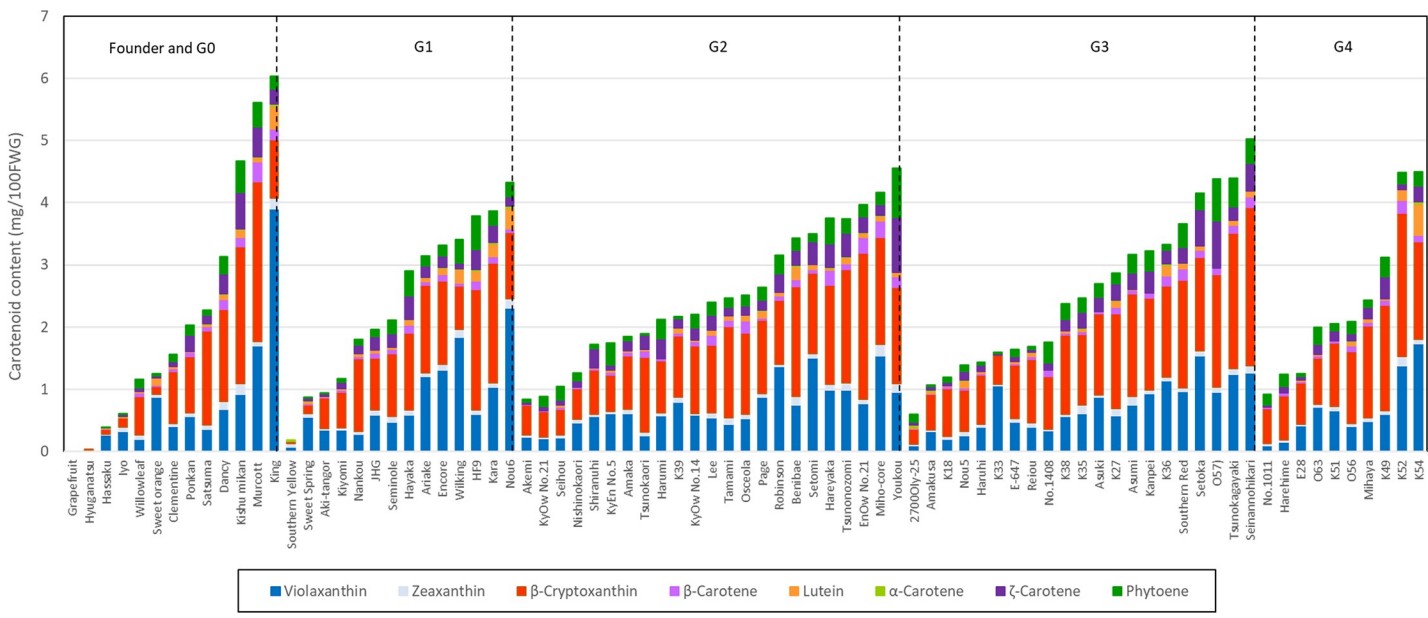

**Fig 3. Carotenoid composition in juice sac tissues of fruit among 13 founders and 57 cultivars and lines within 5th generation. Notes:** Phytoene, α-carotene, β-carotene, ζ-carotene, lutein, β-cryptoxanthin, zeaxanthin, and violaxanthin in juice sac tissues were measured by high-pressure liquid chromatography. G0: natural cross, G1: 1st generation, G2: 2nd generation, G3: 3rd generation, G4: 4th generation.

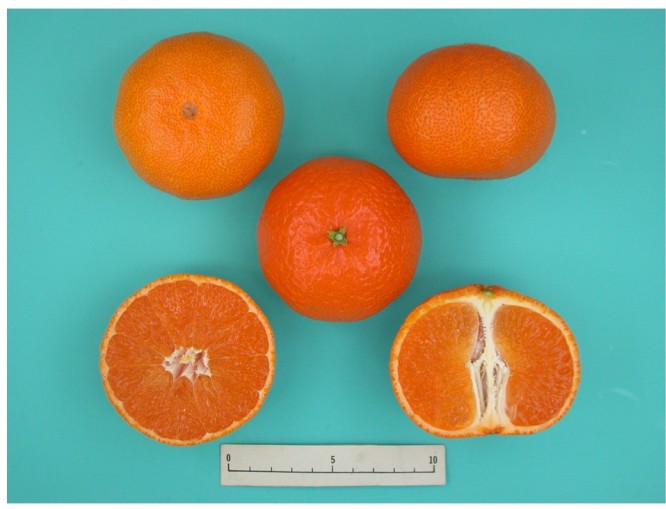

'Tsunokagayaki'

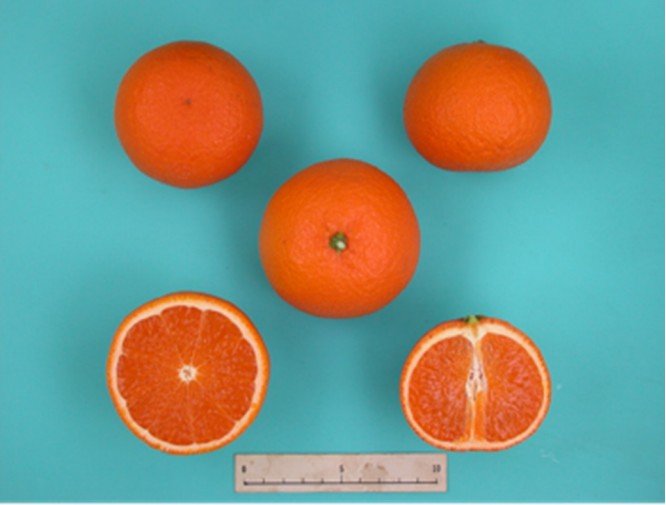

Kuchinotsu 52 gou

**Fig 4. Images of the fruits of 'Tsunokagayaki' and Kuchinotsu 52 gou with *PSY-a* and *ZEP-e*. Notes:** The deep orange color in flesh color revealed high concentration of β-cryptoxanthin content in juice sac tissues of 'Tsunokagayaki' (2.2 mg/100FWG) and Kuchinotsu 52 gou (2.4 mg/100FWG).

carotenoids, β-cryptoxanthin, and VIO became more abundant owing to the removal of unnecessary alleles. For example, 'Tsunokagayaki' in G3 and Kuchinotsu 52 gou in G4 revealed a deep orange flesh color (Fig 4), and β-cryptoxanthin (total carotenoids) content in the juice sac tissues were 2.2 mg/100FWG (4.6 mg/100FWG) and 2.4 mg/100FWG (4.5 mg/100FWG), which were higher than 1.5 mg/100FWG (2.3 mg/100FWG) of Satsuma mandarin in G0. Interestingly, β-cryptoxanthin content and violaxanthin content were highly correlated with the total carotenoid content among breeding populations, and both of their correlation values were 0.85 (Fig 5). This result implied that the increase in total carotenoids was supported by the accumulation of these xanthophylls in mature fruits. These carotenoid profiles suggest that carotenoid enrichment was achieved by the exchange of allelic composition and by the targeted removal of the unnecessary alleles in conventional crossbreeding.

### Association between the allelic composition of five target genes and carotenoid composition in juice sac tissues of fruits by Bayesian statistical analysis

The association between allelic composition of five target genes and carotenoid composition (S3 Table) was evaluated by Bayesian statistical analysis using the data obtained from the 263 pedigrees. The five target genes detected some significant associations with the measured eight kinds of carotenoids and total carotenoid content (Fig 6). *PSY*, *ZEP*, and *NCED* significantly associated with total carotenoid content, *PSY* and *ZEP* significantly associated with β-cryptoxanthin content, and *PSY*, *HYb*, *ZEP*, *NCED*, and *TCL* significantly associated with violaxanthin content. Based on the estimated effect size for each allele of the five target genes on the measured carotenoids, the allele with the value of more than one third of the max effect size was defined as the alleles with positive or negative effect (Fig 7). With respect to the independent alleles of the five target genes, *PSY-a* had a strong positive effect ($\zeta = 0.82$) on increasing the total carotenoid content in juice sac tissues of fruits, whereas *PSY-c* and *PSY-g* had medium negative effects ($\zeta = -0.42$ and $-0.41$, respectively) on decreasing the total carotenoid content

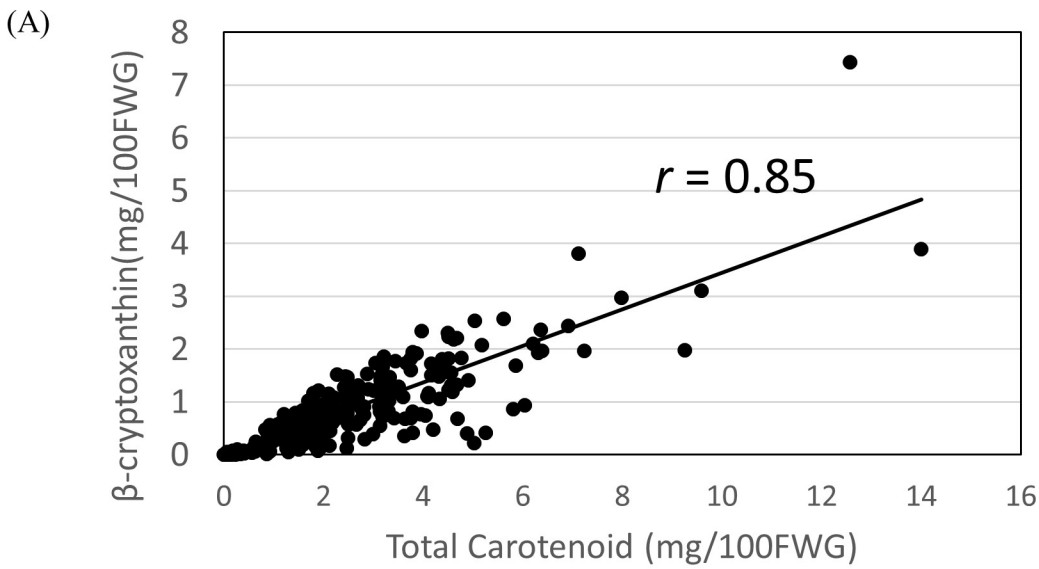

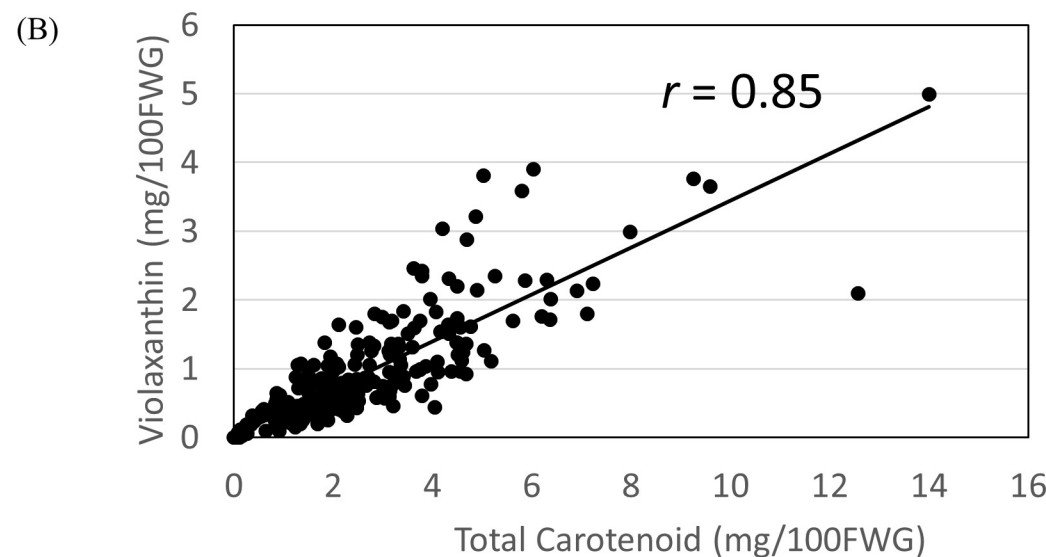

**Fig 5.** Correlation between total carotenoid content and β-cryptoxanthin content (A) and between total carotenoid content and violaxanthin content (B) among 263 breeding pedigrees. **Notes:** β-cryptoxanthin and violaxanthin are major xanthophylls that accumulate in mature fruits, and their increase leads to an increase in total carotenoids.

(Fig 7A). Among the *ZEP* alleles, *ZEP-e* had a strong positive effect ($\zeta = 0.83$) on increasing the total carotenoids in juice sac tissues of fruits, whereas three alleles, *ZEP-a*, *ZEP-b*, and *ZEP-c*, had medium or weak positive effects ($\zeta = 0.54$, $0.38$, and $0.66$, respectively) on increasing the total carotenoid content. Four alleles of *ZEP-f*, *ZEP-g*, *ZEP-h*, and *ZEP-k* had medium or weak negative effects ($\zeta = -0.42$, $-0.68$, $-0.59$, and $-0.31$, respectively) on decreasing the total carotenoid content in juice sac tissues of fruits. Moreover, among the *NCED* alleles, *NCED-a* had a weak positive effect ($\zeta = 0.34$) on increasing the total carotenoid content in juice sac tissues of fruits. In contrast to those genes, *HYb* and *TCL* had unclear effect on the total carotenoid

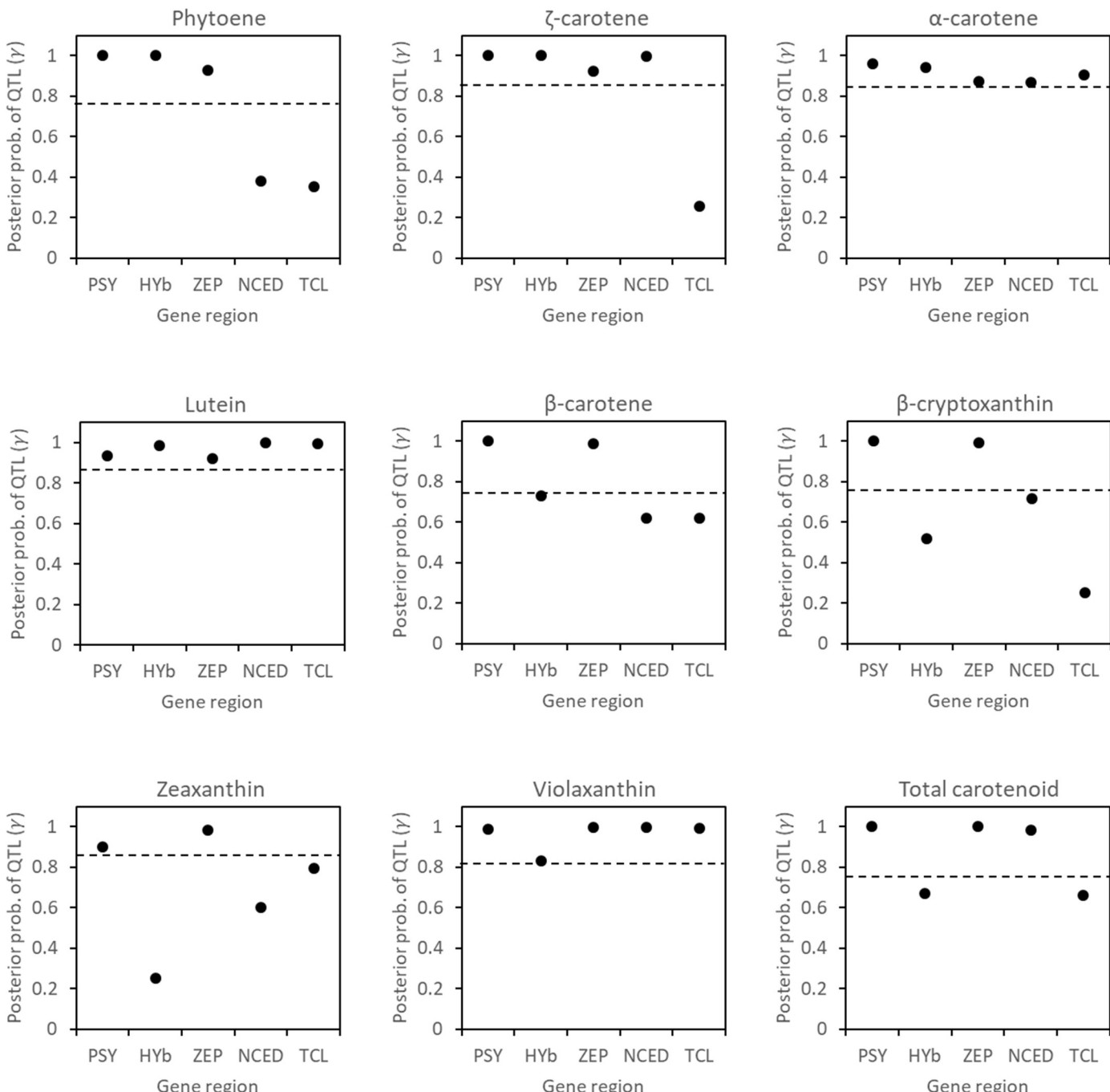

**Fig 6. Posterior probability of having a QTL (posterior average of γ) at 5 targeted gene regions, estimated for each carotenoid.** Notes: Horizontal dashed lines correspond to the threshold obtained from the random permutation procedure.

content in juice sac tissues of fruits. With regard to the β-cryptoxanthin content, *PSY-a* had a medium positive effect ($\zeta = 0.38$), while *PSY-c* and *PSY-g* had weak negative effects ($\zeta = -0.29$ and -0.26, respectively). In contrast, *ZEP-a* had a weak positive effect ($\zeta = 0.25$). Other alleles had unclear effect on β-cryptoxanthin content in juice sac tissues of fruits. Finally, with regard to the violaxanthin content, *PSY-a* had a weak positive effect ($\zeta = 0.20$), while *ZEP-c* and *ZEP-e*

(A)

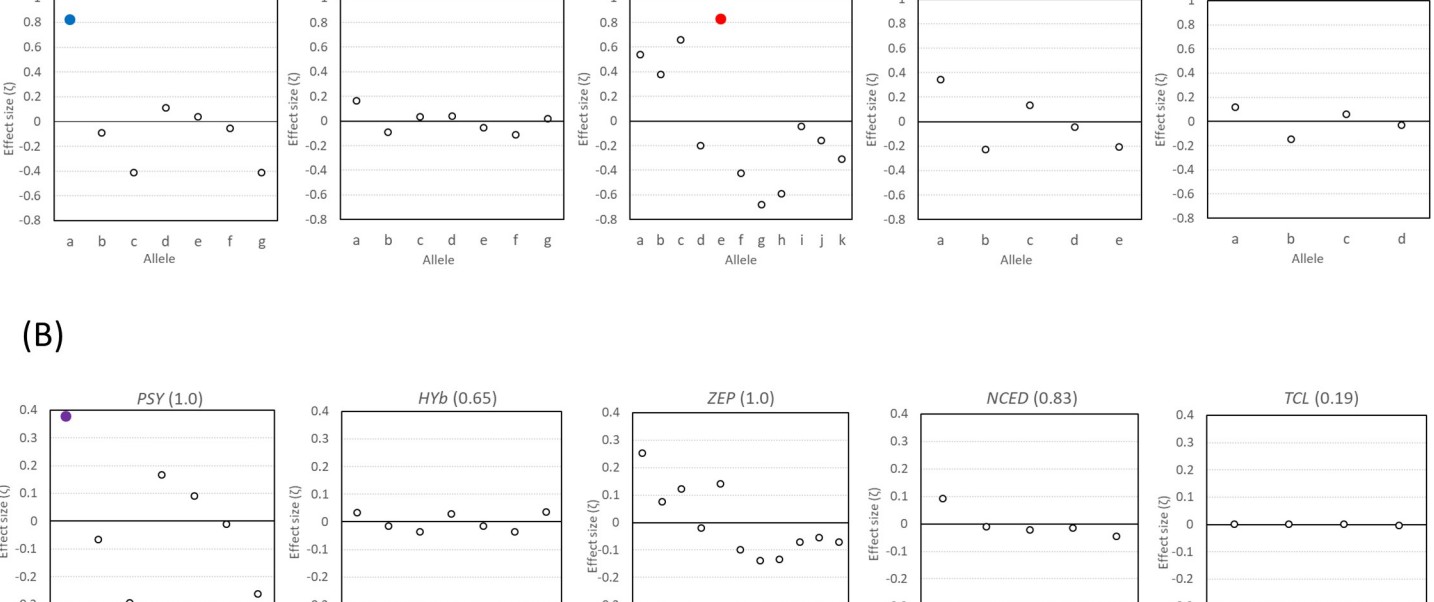

(B)

(C)

**Fig 7.** Estimated effect size ($\zeta$) of each allele in the five target gene regions for (A) total carotenoid, (B) β-cryptoxanthin, and (C) violaxanthin. **Notes:** Red, blue, and purple circle indicated that the allele among the 5 target regions had the highest effect size, the largest variance, and both the highest effect size and the largest variance, respectively. The characters in the brackets mean the product of the posterior means of γ.

also had weak positive effects ($\zeta$ = 0.24 and 0.34, respectively). Other alleles had unclear effect on violaxanthin content in juice sac tissues of fruits.

Regarding the other carotenoids, several alleles in *PSY*, *HYb*, and *ZEP* had a weak effect on increasing their content (S2 Fig). Notably, *PSY-a* had a broad effect on multiple carotenoids, except for lutein.

Fig 8 shows the transmission of *PSY* and *ZEP* alleles in the lineage of 'Tsunokagayaki' in G3 and Kuchinotsu 52 gou in G4, which accumulate high concentrations of β-cryptoxanthin in juice sac tissues. The alleles of *PSY* and *ZEP* come from either allele in five founders of Kishu

(A)

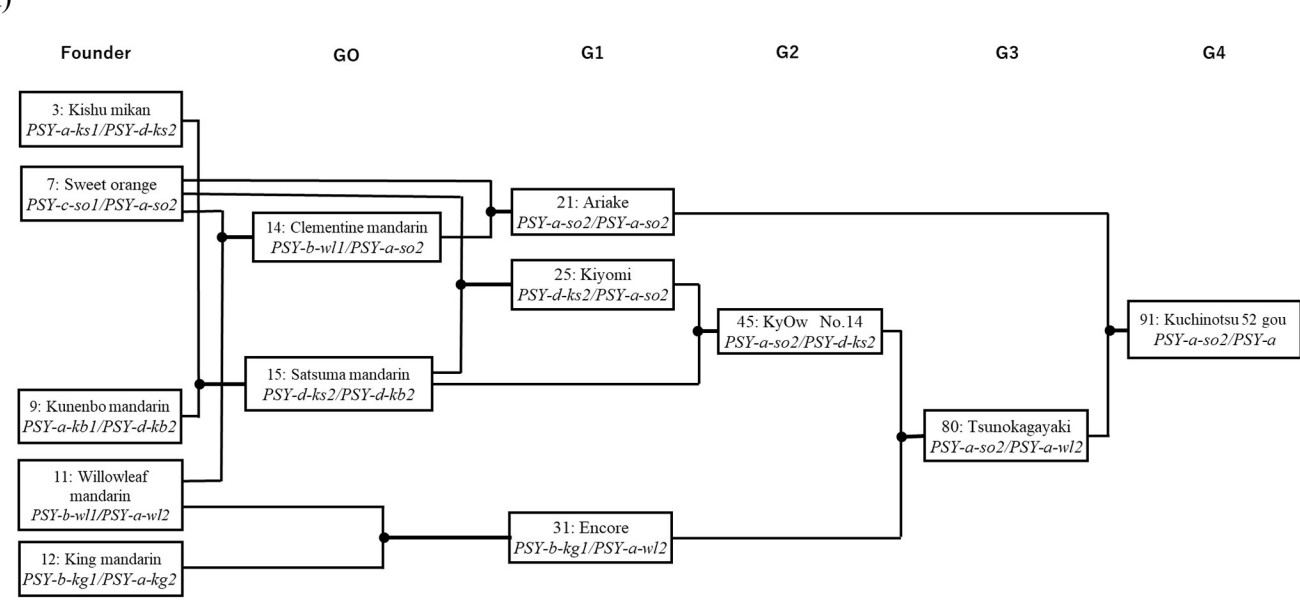

(B)

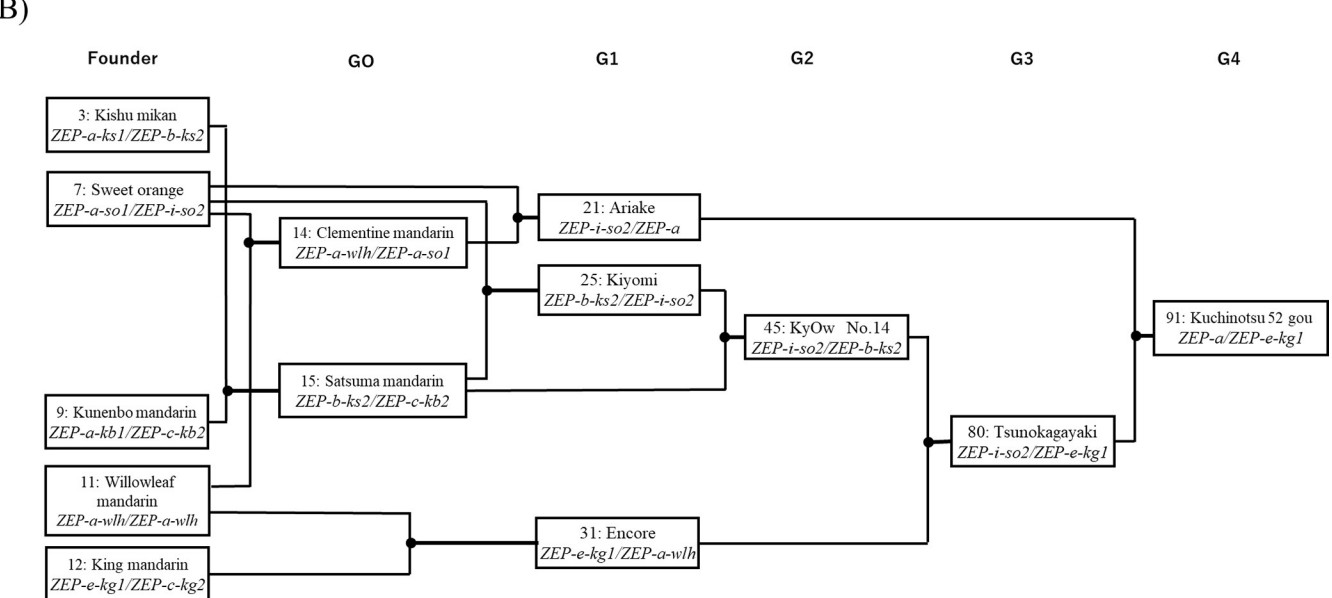

**Fig 8. Transmission of *PSY* and *ZEP* alleles in the lineage of 'Tsunokagayaki' and Kuchinotsu 52 gou with high concertation of β-cryptoxanthin. Notes:**
G0: natural cross, G1: 1st generation, G2: 2nd generation, G3: 3rd generation, G4: 4th generation. *PSY* (*PSY-a*) and *ZEP* (*ZEP-a* and *ZEP-e*) alleles with positive effects are selected in the process of conventional breeding program for the enrichment of β-cryptoxanthin.

mikan (*PSY-a*/*PSY-d*; *ZEP-a*/*ZEP-b*), sweet orange (*PSY-a*/*PSY-c*; *ZEP-a*/*ZEP-i*), Kunenbo mandarin (*PSY-a*/*PSY-d*; *ZEP-a*/*ZEP-c*), willow leaf mandarin (*PSY-a*/*PSY-b*; *ZEP-a*/*ZEP-a*), and King mandarin (*PSY-a*/*PSY-b*; *ZEP-c*/*ZEP-e*). During breeding selection aiming for enrichment of β-cryptoxanthin, *PSY-a* and *ZEP-e* (or *ZEP-a*) were pyramided in those

cultivars and breeding lines, thus confirming that *PSY-a* and *ZEP-e* had statically strong positive effects on the increase in total carotenoid content in juice sac tissues of fruits. These alleles would operate to increase β-cryptoxanthin and violaxanthin contents, although the effects were smaller compared to those exerted overall on the total carotenoid content. Of note, most alleles with negative effects on carotenoid content were, consistently, those alleles that disappeared during the progress of breeding generation.

## Verification of the effects of optimum *PSY* and *ZEP* alleles on increased carotenoid content in juice sac tissues of fruits

Box plot analysis was carried out to inspect the ability of *PSY* and *ZEP* optimum alleles with the strongest positive effects on total carotenoid content in juice sac tissues of fruits. Fig 9A revealed the distribution of total carotenoid content in six combinations of *PSY* allele as follows: condition 1: a pair of negative alleles (*PSY-c* or *PSY-g*), condition 2: a single of negative alleles without an optimum allele *(PSY-a)*, condition 3: not including both optimum (*PSY-a*) and negative alleles (*PSY-c* or *PSY-g*), condition 4: a single optimum allele (*PSY-a*) and a single negative allele (*PSY-c* or *PSY-g*), condition 5: a single optimum allele (*PSY-a*) without any negative allele (*PSY-c* or *PSY-g*); and condition 6, 2 optimum alleles (*PSY-a*).When the mean values between conditions 2 and 3 were compared, it was clear that *PSY-c* and *PSY-g* had negative effects on reducing the carotenoid concentration. In contrast, the mean value of condition 6 with two *PSY-a* alleles was 4.2 mg/100FWG, which was higher than those of conditions 3, 4, and 5. Interestingly, the allelic interaction between alleles with positive effect and negative effect would be observed in condition 4

In addition, the ability of 4 *ZEP* alleles (*ZEP-a*, *ZEP-b*, *ZEP-c*, and *ZEP-e*), which were identified to exert a positive effect on the total carotenoid content, was examined against the above six conditions of *PSY* alleles (Fig 9B). Under condition 3, which did not include any *PSY-a*, the mean values of total carotenoid content increased proportionately with the number of *ZEP a*lleles. Under conditions 1 and 2, no consistent effect was observed along with an increase in the number of *ZEP* alleles, indicating an epistatic interaction between the *PSY* allele and *ZEP* allele. This result was understandable as *PSY* is located upstream of the carotenoid metabolic pathway compared to *ZEP*. Consequently, *PSY* alleles with negative effects could negate the effectiveness of *ZEP* alleles with positive effects. Under condition 5, the mean values of total carotenoid content increased proportionately with the number of optimum *ZEP* alleles. Under condition 6, the mean values of total carotenoid content with those *ZEP* alleles were higher than those without them. Thus, these results confirmed that the optimum alleles of *PSY* and *ZEP* possess the capability of increasing carotenoid content in juice sac tissues of fruits.

Box plot analysis was also carried out for β-cryptoxanthin and violaxanthin content (S3 and S4 Figs, respectively). With respect to β-cryptoxanthin, a slight effect of *PSY-a* was observed on increasing the mean value, while *ZEP* alleles with a positive effect clearly indicated that their numerical increase led to an increase in β-cryptoxanthin content. Furthermore, for violaxanthin, the increase in *PSY-a* led to an increase in its mean value. In contrast, the numerical increase of *ZEP* alleles with positive effects did not correlate with violaxanthin content, except for condition 5. These evaluation test results indicated that the optimum allele of *PSY-a* emphatically had the ability to strengthen the flux of carotenoid metabolic pathway and increase total carotenoid content, along with an increase in β-cryptoxanthin and violaxanthin. *ZEP* alleles with a positive effect also confirmed an increase in total carotenoids, and its effect preferentially increased β-cryptoxanthin content over violaxanthin content. It is considered that *ZEP* alleles with positive alleles did not have a strong effect on increasing β-cryptoxanthin content by themselves, and they would likely synergistically increase β-cryptoxanthin content

(A)

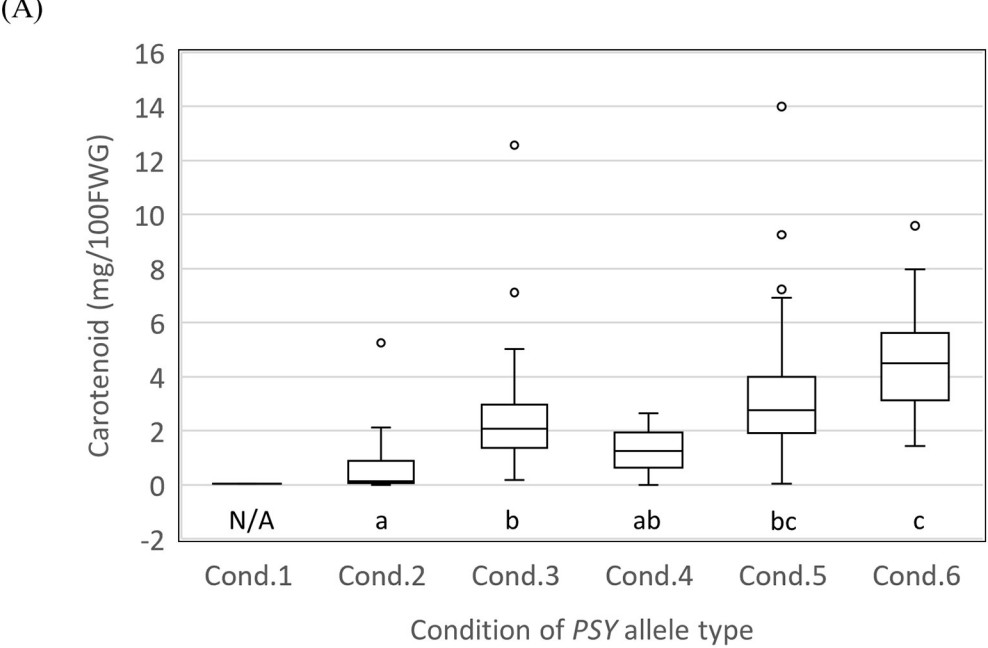

(B)

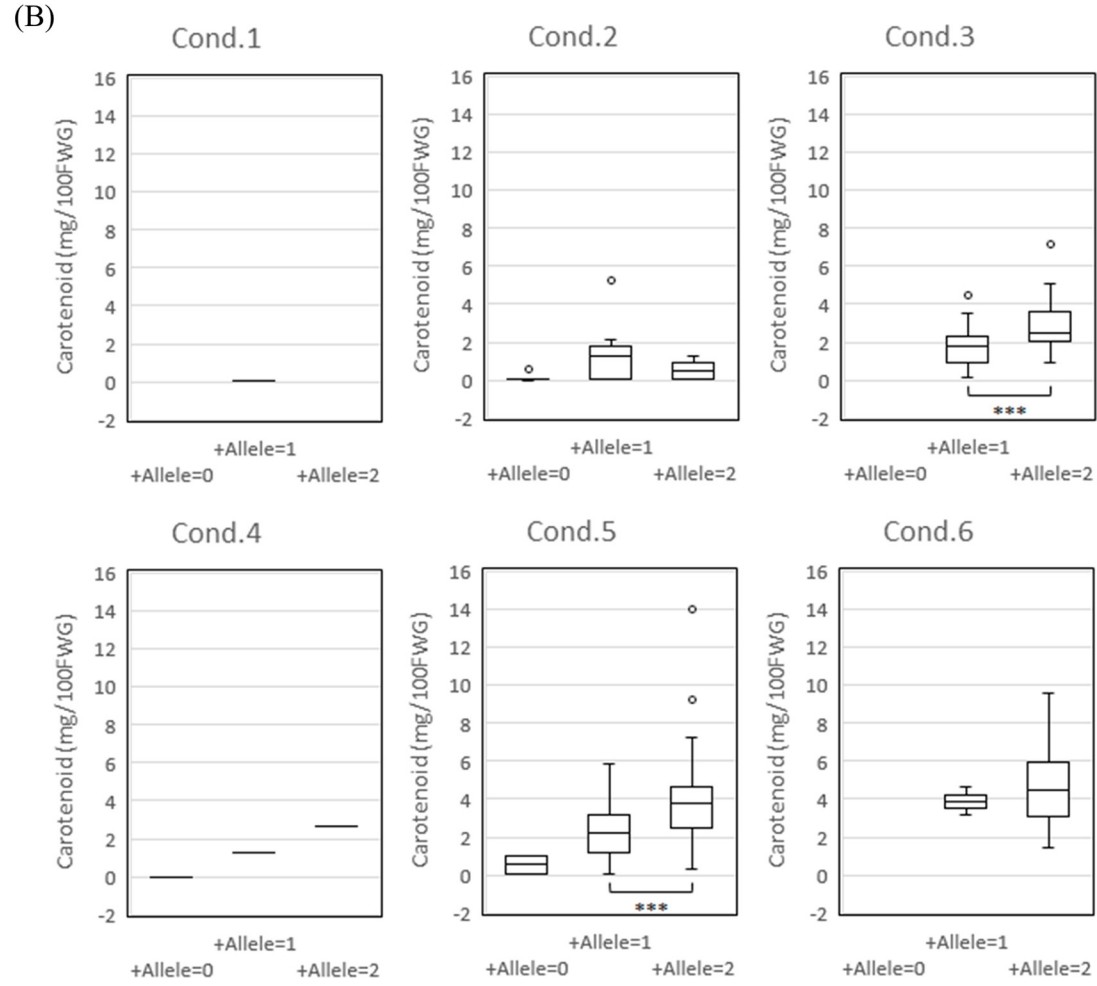

**Fig 9. Box plot analysis to verify the effectiveness of optimum *PSY* and *ZEP* alleles for the enrichment of total carotenoids in juice sac tissues of fruits. Notes:** Box edges represent the upper and lower quantiles, with the median value shown as a bold line in the middle of the box. The circles indicate an outlier. (A) Comparisons of total carotenoid contents by the number of alleles with positive effect (*PSY-a*) and negative effect (*PSY-c* or *-g*). Condition 1, *PSY-a* = 0 & (*PSY-c* or *-g*) = 2 Condition 2, *PSY-a* = 0 & (*PSY-c* or *-g*) = 1; Condition 3, *PSY-a* = 0 & (*PSY-c* or *-g*) = 0; Condition 4, *PSY-a* = 1 & (*PSY-c* or *-g*) = 1; Condition 5, *PSY-a* = 1 & (*PSY-c* or *-g*) = 0; Condition 6, *PSY-a* = 2 & (*PSY-c* or *-g*) = 0. Means were tested for significant differences using Tukey's HSD test at P < 0.05, following the normality of the distribution was confirmed not to be rejected at *P* = 0.05 by Kolmogorov-Smirnov's one sample test. Carotenoid contents were analyzed as square root-transformed values. N/A indicates entry eliminated from the Tukey's HSD test due to insufficient data. (B) Comparison of total carotenoid content by the number of alleles with a positive effect (*ZEP-a, b, c,* or *e*) under the PSY allele-type conditions (A). Means between '+Allele = 1' and '+ Allele = 2' were tested for significant differences using Welch's two-sample t-test under conditions 3 and 5. *** indicates significance at P < 0.001, following the normality of the distribution was confirmed not to be rejected at *P* = 0.05 by Kolmogorov-Smirnov's one sample test. Carotenoid contents were analyzed as square root-transformed values. Means under other conditions were eliminated from Welch's two-sample t-test due to insufficient data.

in cooperation with *PSY-a* by the sandwich effect between upstream and downstream genes in the carotenoid pathway.

## Minimal set of trio-tagged SNPs to identify the optimum alleles of *PSY-a* and *ZEP-e* for marker-assisted selection

Pyramiding for optimum alleles of *PSY-a* and *ZEP-e* would be promising for the enrichment of carotenoids, especially β-cryptoxanthin in juice sac tissues of fruits. To efficiently promote molecular breeding by marker-assisted selection (MAS), minimal sets of trio-tagged SNPs were evaluated using MinimalMarker software [35] in order to determine the optimum *PSY* and *ZEP* alleles among the others. For *PSY*, MinimalMarker software revealed 45 minimal sets, which comprised different combinations of six trio-tagged SNPs, to discriminate any combination of the seven *PSY* alleles. For example, PSY-SNP-05, PSY-SNP06, PSY-07, PSY-SNP08, PSY-SNP09, and PSY-SNP10 were one of the minimal sets to discriminate each other (Table 3). Fortunately, the optimum *PSY-a* allele could be discriminated with the other alleles by a single trio-tagged PSY-SNP06. For *ZEP*, MinimalMarker software revealed a single minimal set comprising seven trio-tagged SNPs (ZEP-SNP01, ZEP-SNP03, ZEP-SNP05, ZEP-SNP14, ZEP-SNP17, ZEP-SNP20, and ZEP-SNP21) to discriminate each other (Table 4). The optimum *ZEP-e* allele could be discriminated with the other alleles by two trio-tagged ZEP-SNP17 and ZEP-SNP21. When the allelic genotypes for *PSY* and *ZEP* alleles in the parent varieties are available, the number of SNPs for MAS in their progenies will be smaller than the above calculated minimal sets. For example, the genotypes of seedlings from the cross between

**Table 3. Minimal set of trio-tagged SNPs that discriminate 7 *PSY* independent alleles in 13 founders.**

| Independent allele | Trio-tagged SNPs | | | | | |
|---|---|---|---|---|---|---|
| | PSY-SNP05 | PSY-SNP06 | PSY-SNP07 | PSY-SNP08 | PSY-SNP09 | PSY-SNP10 |
| *PSY-a* | T | A | A | A | C | T |
| *PSY-b* | T | G | A | G | C | T |
| *PSY-c* | T | G | A | G | C | C |
| *PSY-d* | C | G | A | A | C | T |
| *PSY-e* | T | G | A | A | T | T |
| *PSY-f* | T | G | C | A | C | T |
| *PSY-g* | T | G | A | A | C | T |

**Notes:** *PSY*, phytoene synthase gene. 7 PSY-independent alleles were named from *PSY-a* to *PSY-g*. The all trio-tagged SNP genotypes for *PSY* are summarized in S2 Table.

**Table 4. Minimal set of trio-tagged SNPs that discriminate 11 *ZEP* independent alleles in 13 founders.**

| Independent allele | Trio-tagged SNPs | | | | | | |
|---|---|---|---|---|---|---|---|
| | ZEP-SNP01 | ZEP-SNP03 | ZEP-SNP05 | ZEP-SNP14 | ZEP-SNP17 | ZEP-SNP20 | ZEP-SNP21 |
| *ZEP-a* | A | C | A | C | T | A | T |
| *ZEP-b* | A | C | A | C | C | A | C |
| *ZEP-c* | A | C | A | C | C | C | C |
| *ZEP-d* | T | C | A | C | C | C | C |
| *ZEP-e* | A | C | A | C | T | A | C |
| *ZEP-f* | A | C | A | C | T | C | T |
| *ZEP-g* | T | A | A | C | C | A | C |
| *ZEP-h* | A | C | A | T | C | C | C |
| *ZEP-i* | T | C | T | C | C | C | C |
| *ZEP-j* | T | A | A | C | C | C | C |
| *ZEP-k* | A | C | T | C | C | C | C |

**Notes:** *ZEP*, zeaxanthin epoxidase gene. 11 ZEP-independent alleles were named from *ZEP-a* to *ZEP-k*. The all trio-tagged SNP genotypes for *ZEP* are summarized in S2 Table.

the seed parent with *ZEP-c* and *ZEP-e* and pollen parent with *ZEP-e* and *ZEP-e* would reveal homozygous genotypes of *ZEP-e* or heterozygous genotypes of *ZEP-c* and *ZEP-e*. In this case, only one trio-tagged ZEP-SNP17 or ZEP-SNP20 was sufficient to discriminate the seedlings with homozygous genotypes of *ZEP-e* (Fig 10). Other trio-tagged SNPs, which were not included in the above minimal marker sets, are also applicable for MAS when they discriminated parent allelic genotypes. Recently, various SNP detection systems have been developed, such as the TaqMan-MGB SNP genotyping assay, Kompetitive Allele Specific PCR genotyping assay, and so on. SNP markers compatible with those systems could be customized based on the sequence information of trio-tagged SNPs listed in S1 Table.

## Discussion

The carotenoid profile in citrus fruits varies greatly across species and cultivars, wherein approximately 115 different carotenoids have been reported [4]. To address how carotenoid composition is highly extended among cultivars, we focused on the allelic diversities of carotenoid metabolic genes. Recent advances in NGS technologies provided a new insight that citrus varieties comprise a complex mosaic genome structures of a limited number of ancient species via repeated natural cross [35]. Ahmed et al. [38] further revealed interspecific mosaic genome structures in 53 citrus accessions that comprised large genomic fragments derived from four basic taxa using 15,946 diagnostic SNP markers generated by genotyping by sequencing assay. It is plausible that some of the founders in the Japanese breeding population share the common haplotype block derived from ancient species, where functionally synonymous genes are located. Citrus genomes are highly heterozygous and maintain high sequence variation among citrus varieties, resulting in numerous SNPs that can be detected in sequence comparisons made among them. The high frequency of SNPs among citrus varieties makes it difficult to explore the functional SNPs linked to important agricultural traits. In this study, trio-tagged SNPs, which were inherited in lineage, were used to classify the independent allele of 5 target genes, including a causative gene on *TCL*, in 13 founders. The number of independent alleles in the 13 founders was smaller than expected. This result was supported by previous reports that modern citrus varieties revealed extensive sharing of haplotypes from the ancient species among them [37]; the collapse of linkage disequilibrium (LD) structure was

(A)

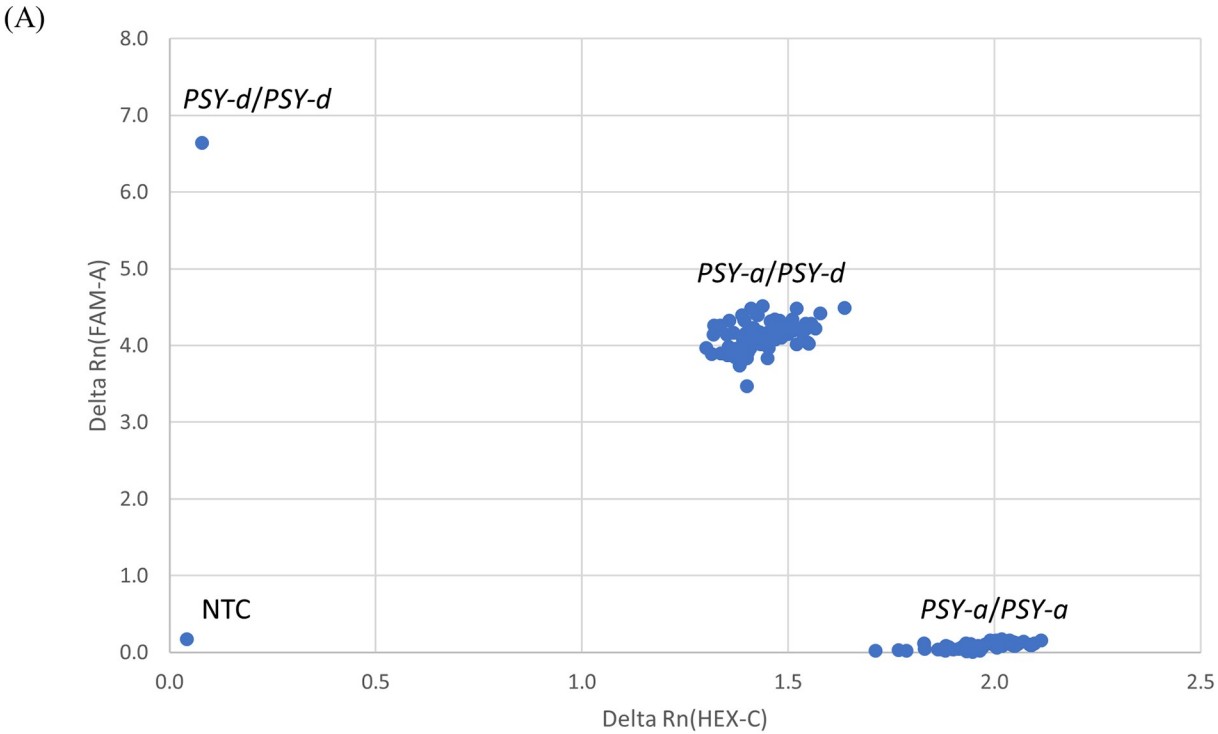

(B)

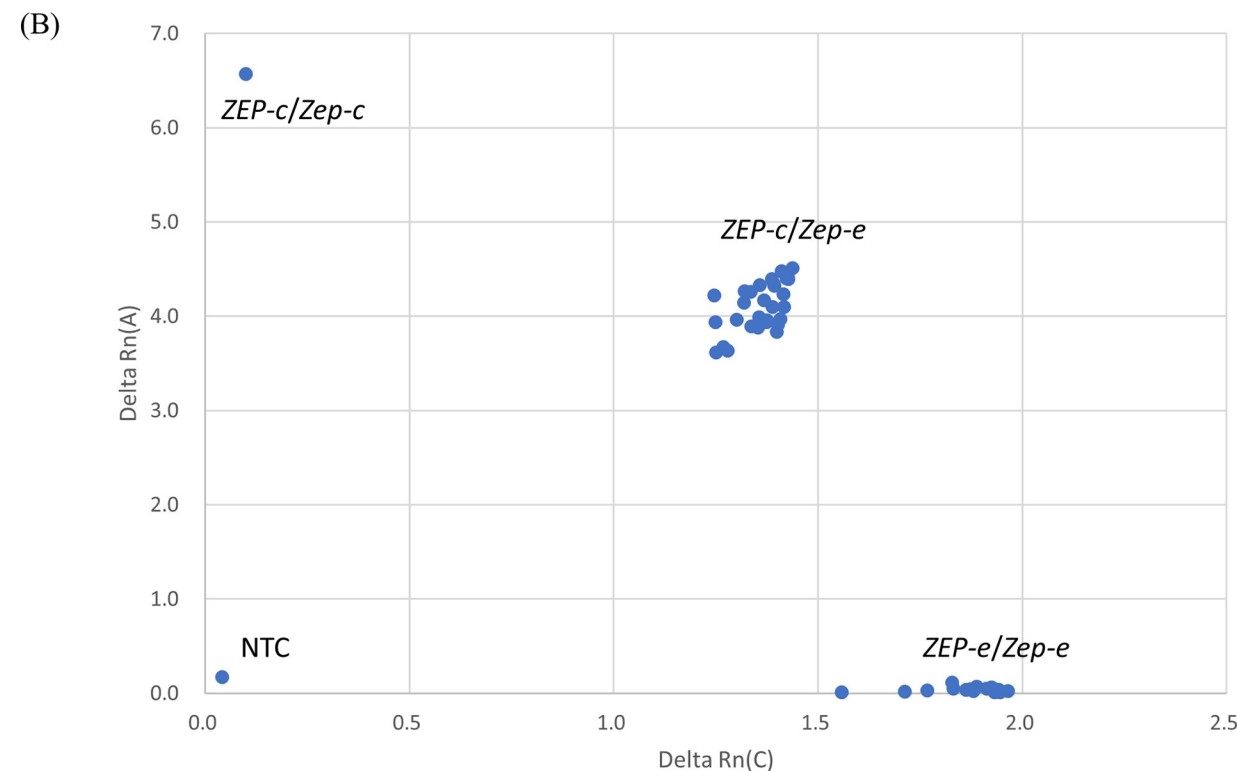

**Fig 10. TaqMAN-MGB SNP genotyping using PSY-SNP06 and ZEP-SNP17 for marker-assisted selection. Notes:** Seedlings with homozygous *PSY-a* (A) and those with homozygous *ZEP-e* (B) are able to select in marker-assisted selection for the enrichment of carotenoids by a single DNA marker when allelic genotypes of parents are available. FAM fluorescent signal values are plotted on the x-axis, and HEX fluorescent signal values are plotted on the y-axis. NTC: Non-template control.

very slow [39], and a wide range of LD was observed in the breeding population [40]. Within the independent alleles of the five target genes, the genotypes of the trio-tagged SNPs were identical among the examined breeding populations. We considered that independent alleles of the five target genes would play synonymous functional roles in carotenoid metabolism because they are located on common haplotype blocks derived from common founders, even if numerous *de novo* mutations are scattered among them.

## *PSY* and *ZEP* play a key role in influencing carotenoid content in juice sac tissues of fruits

Out of the five target genes responsible for high carotenoid content, *PSY-a* and *ZEP-e* were identified as the optimum alleles responsible for increasing carotenoid content in juice sac tissues of fruits. PSY is a rate-limiting enzyme in the carotenoid biosynthetic pathway, and the flux of the carotenoid pathway is generally controlled by isoforms or alleles of *PSYs* in plants [41]. Expression profiles of different *PSY* isoforms are likely to exhibit tissue specificity in rice (*Oryza sativa* L.) and tomato (*Solanum lycopersicum* L.) [42, 43]. In clementine mandarin, there are four *PSY* isoforms, and the locus of Ciclev10011841m.g in scaffold 6 is responsible for carotenoid biosynthesis in juice sac tissues of the fruit [24]. This locus possessed plural *PSY* alleles with sequence variation among citrus varieties, and many expressed sequence tags are registered in the public database. We confirmed that the allelic combination of *PSYs* on this locus influenced *PSY* transcription level and carotenoid content in the fruits of $F_1$ plants from a cross between A255 and G434. One *PSY* allele of G434 had a negative effect on the transcription of *PSY*, and $F_1$ plants with this allele tended to have lower carotenoids than those without it. Several reports have documented that an increase in the carotenoid pathway flux emphasized the carotenoid accumulation by overexpression of *PSY* in tomato, *Arabidopsis*, and white carrots (*Daucus carota* L) [44, 45]. Therefore, the flux of the carotenoid pathway is primarily controlled by the allelic combination of seven *PSY* alleles in the Japanese breeding population. Among these, *PSY-a* had a highly positive effect on increasing the total carotenoid content, while *PSY-c* and *PSY-g* had a negative effect on decreasing it. The *PSY* independent alleles in 13 founders were arranged based on the 12 trio-tagged SNPs. Interestingly, phylogenetic tree analysis revealed that the seven *PSY* alleles functionally diverged from the common *PSY* ancestor during the process of citrus species specification (S5 Fig). All trio-tagged SNPs located on the out of open reading frames and those SNPs were nor likely linked to influence enzymatic activity. *PSY-a* also had a positive effect on β-cryptoxanthin and violaxanthin accumulation, but the effects were weaker compared to those exerted overall on the total carotenoids. In citrus fruits, the major xanthophyll in the juice sac is either β-cryptoxanthin or violaxanthin, and its content increases during fruit maturation. We considered that the high positive effect of *PSY-a* on total carotenoids would be due to the sum of these effects on β-cryptoxanthin and violaxanthin.

ZEP metabolizes zeaxanthin to yield antheraxanthin and then violaxanthin. Violaxanthin is the substrate for the synthesis of neoxanthin and can be converted back to zeaxanthin by violaxanthin deepoxidase. This reversible epoxidation/deepoxidation is termed the xanthophyll cycle, in which deepoxidation to zeaxanthin is favored under high-light conditions, while epoxidation to violaxanthin predominates in moderate-light conditions. The rapid formation

of zeaxanthin by the xanthophyll cycle is integral for the dissipation of excess energy by non-photochemical quenching [46]. *ZEP-e* had a strong effect on increasing the total carotenoid content, while *ZEP-a*, *ZEP-b*, and *ZEP-c* had medium or weak effects on it. In contrast, *ZEP-g* and *ZEP-h* had a negative effect on decreasing the total carotenoid content. The *ZEP* independent alleles in 13 founders were arranged based on the 24 trio-tagged SNPs. Phylogenetic tree analysis revealed that the *ZEP* alleles with positive effects (*ZEP-a*, *ZEP-b*, *ZEP-c* and *ZEP-e*) were functionally diverged from negative allele group, and *ZEP-f* might be further diverged from positive allele group (S6 Fig). Eight trio-tagged SNPs located on the open reading frames but the amino acid substitution that linked to influence enzymatic activity was not found between positive and negative alleles. In carrot, maize, and potato (*Solanum tuberosum* L.) [47–49], *ZEP* was identified as a major candidate gene governing carotenoid accumulation. A lower expression level of *ZEP* resulted in the accumulation of zeaxanthin and upstream of carotenoids in these crops. Homozygosity of the recessive *ZEP* allele, with the insertion of a non-LTR retrotransposon in the promoter region, caused a reduced level of expression and could accumulate large amounts of zeaxanthin in potato. In citrus, Sugiyama et al. [23] reported that the *ZEP* allelic combination influenced the amounts and accumulation patterns of their transcripts, altering the carotenoid content and composition in juice sac tissues of fruits. The high transcription allele of *ZEP-1o* in the previous report was *ZEP-g*, with a negative effect in this study, whereas the low transcription alleles of *ZEP-1m*, *ZEP-2m*, and *ZEP-2o* in the previous report were *ZEP-a*, *c*, and *e*, respectively, with a positive effect in this study. Considering that *ZEP* is located downstream of the carotenoid biosynthesis pathway, *ZEP* alleles with positive effects would narrow the flux of the carotenoid pathway and increase the carotenoid content in juice sac tissues of fruits.

In this study, a weak positive effect of *NCED-a* was detected against total carotenoid content, but any effective allele was not detected on β-cryptoxanthin and violaxanthin. NCED is encoded by a multigene family, and stress-inducible members contribute to a rate-limiting step in abscisic acid (ABA) biosynthesis [50]. The functional characterization of NCED in response to water deficit has been well characterized in the vegetative tissues, whereas the contribution of other NCED family members appeared to be minor. In citrus, NCED has been characterized at the molecular level in fruits and leaves [51, 52]. They demonstrated a reciprocal relationship between the levels of *NCED* transcripts and 9-*cis*- violaxanthin (a substrate of NCED), suggesting that NCED activity affected carotenoid accumulation in citrus fruits via ABA synthesis. The oxidative cleavage of 9-*cis*-violaxanthin catalyzed by NCEDs affects the concentration of 9-*cis*-violaxanthin and ABA in fruits, and consequently the β, β-xanthophyll composition [11]. Ma et al. [53] reported that the low transcription level of *NCED* in the mature fruit played a key role in increasing β-cryptoxanthin and violaxanthin content in the mature fruit when comparing Satsuma mandarin and 'Seinannohikari' (EnOw No.21 × 'Yoko'). NCED is also located at the end of the carotenoid metabolism pathway, and its allelic genotype also influences carotenoid accumulation. Because 'Seinannohikari' does not have optimum allele of *PSY-a* and *ZEP-e* (S3 Table), NCED would like to play a visible role to increase β-cryptoxanthin and violaxanthin content instead of them.

Thus, it was found that *PSY* and *ZEP* play key roles in influencing the carotenoid content in juice sac tissues of fruits. By pyramiding of *PSY-a* and *ZEP-e* alleles located at the beginning and end of the carotenoid pathway, a significant increase in carotenoid concentration could be expected in juice sac tissues of fruits. In this study, we could not detect the effects of *HYb* and *TCL*. As these genes are located in the middle of the carotenoid biosynthetic pathway, their effects may be hidden by the epistatic interaction between upstream and downstream allelic genotypes.

## Other factors influencing carotenoid accumulation in citrus mature fruit

In this study, we provided a new insight that the carotenoid composition in fruit was highly influenced by the allelic composition of carotenoid metabolic genes. The expected variance explained by five target genes for phenotypic variance of total carotenoid content among 263 breeding pedigrees was approximately 0.3, which was similar to that of QTLs for total carotenoids and β-cryptoxanthin detected in bi-parental populations [17]. We considered this to be a reasonable value because the carotenoid pathway was highly influenced by its response to numerous factors, such as development, ABA, high light, salt, drought, and pathogen interaction temperature. In addition, a significant number of terpene synthase (TPS) genes are found in the citrus genome, which are responsible for various terpenoid products such as aroma and limonoid. Because these TPS genes share an isopentenyl precursor of isopentenyl pyrophosphate, the genetic composition of other TPS genes may influence carotenoid accumulation in fruits.

Recently, it has also become clear that several factors influence carotenoid accumulation in mature citrus fruits. Carotenoids are catabolized enzymatically by a family of carotenoid cleavage dioxygenases (CCDs). The *Arabidopsis* genome has nine CCD family genes, which are divided into two groups of CCDs (4 *CCD* genes) and NCEDs (5 *NCED* genes) that recognize different carotenoid substrates and are cleaved at different sites, producing various apocarotenoids, such as precursors of phytohormones (ABA and strigoractone), volatile compounds (β-ionone and related compound), pigments (β-citraurin and bixin), and so on [54]. In tomato, petunia (*Petunia×hybrida*), grape (*Vitis vinifera*), and peach (*Prunus persica*), the transcription of *CCD1* and *CCD4* is associated with the emission of carotenoid-derived volatile scent and pigmentation in flowers and fruit flesh [41]. Carotenoid accumulation in various plant species is generally found to be negatively correlated with the transcription of *CCD1* and *CCD4*. In citrus, the clementine mandarin genome possesses 14 CCD family genes, which can be functionally divided into five subfamilies, namely, *CCD1*, *CCD4*, *CCD7*, *CCD8*, and *NCED* [55]. Citrus CCD1 has wide substrate specificity and generates carotenoid-derived volatile scent [51], whereas CCD4 generates β-citraurin from β-cryptoxanthin and zeaxanthin, which is responsible for the red pigmentation of citrus fruits [56]. There are several reports that *CCD1* and *CCD4* transcription do not correlate with carotenoid accumulation in morning glory and citrus [51, 57]. A recent recombinant protein assay of citrus CCD1 and CCD4 revealed that they could cleave the free β-cryptoxanthin but not the β-cryptoxanthin esters, indicating that the β-cryptoxanthin esters may be more stable than free β-cryptoxanthin in citrus fruits [58]. Furthermore, more than 80% of the β-cryptoxanthin in citrus fruits is esterified with fatty acids such as laurate, myristate, and palmitate. Therefore, esterification is an important process for the massive accumulation of carotenoids in chromoplasts, as esterification facilitates sequestration and enhances the stability of carotenoids in chromoplasts [59]. Xanthophyll esterase (XES) is a key enzyme responsible for the high carotenoid accumulation in the petals of tomato and petunia (*Petunia × hybrida*) [59, 60]. Therefore, it is possible that XES may also be involved in the regulation of carotenoid accumulation in citrus fruits.

In addition to the CCD family, orange protein (OR) is another factor that influences carotenoid accumulation in the mature fruit. It has been reported that OR increases carotenoid accumulation in various plant species through post-transcriptionally regulating *PSY*, promoting formation of carotenoid-sequestering structures, and preventing carotenoid degradation [61]. Carotenoids are abundant in the chromoplast, and chromoplast formation during fruit ripening is one of the main factors governing fruit pigmentation [62]. In citrus, little is known about the molecular function of OR; thus, it is necessary to investigate the interaction between the high concentration allele of *PSY-a* and the putative citrus OR. In addition, various

transcription factors of *CubHLH1* [14] (Endo *et al.* 2015), an R2R3-MYB transcription factor *CrMYB68* [15] as well as *CsMADS6* [16] have been implicated in the direct regulation of multiple genes involved in carotenoid metabolism during fruit development and maturation. Considering that their putative loci, which were predicted using the genome sequence in MiGD, were not associated with the previously detected QTL [17], these transcription factors may also have some effects on the regulation of carotenoid accumulation, although they prefer to coordinate transcription in response to environmental factors such as light and temperature during fruit development and maturation.

## Allelic mining system aiming to enrich carotenoid concentration in fruit

To date, only a bi-parental population has been used to explore the genomic region responsible for high carotenoid content in juice sac tissues of fruits by QTL and eQTL analyses [17, 20]. The information from this approach would be limited when considering the abundance of the genes and their alleles among the 13 founders. In fact, the offset combination of alleles with positive and negative effects and the epistasis among metabolic genes confuse us to explore the alleles responsible for the enrichment of carotenoid in fruit. In *Arabidopsis*, tomato, maize, and rice, a multi-parental integrated population has been developed to increase the accuracy of genetic studies and lead to the successful elucidation of complex QTLs that are regulated by multiple genes, responsible for the transition from vegetative to reproductive stage, yield, grain quality, response to abiotic and biotic stress, and so on [63–66]. The Japanese citrus breeding population comprises a similar structure of a multi-parental intermated population. Therefore, we hypothesized that association analysis focusing on the alleles of carotenoid metabolic genes would provide more accurate information that is applicable to whole breeding populations rather than QTL analysis using only bi-parental populations.

Recently, genome-wide association studies (GWAS) and genomic selection (GS) using NGS-based genome-wide markers have been widely used in genetic dissection of complex traits in various plants. In *Arabidopsis* and rice [67, 68], GWAS has been applied to find genotype-metabolite associations, providing deeper insights into the genetic basis of metabolic diversity. In citrus, it was confirmed that GWAS and GS revealed the effectiveness of genetic improvement of 17 fruit traits, including fruit weight, smoothness of pericarp, and sugar content [40]. GWAS is a powerful method to determine genotype-phenotype associations; however, the genetic interaction between epistasis or between loci and the environment sometimes prevents the detection of valid associations. In fact, epistatic interactions among carotenoid metabolic genes were also found in this study. In this regard, the developed allelic mining system for carotenoid enrichment would provide a more reliable association between allele composition and carotenoid composition. It is possible that these methods would detect another factor that influences the enrichment of carotenoids in fruit. The new findings obtained in this study are not contradictory to previous physiological and genetic studies.

## Conclusion

We provided a new insight that 13 ancestral varieties of Japanese breeding pedigrees shared common alleles of carotenoid metabolic genes from ancestral species, and the carotenoid diversity observed in cultivars could be extended by exchanging the allelic combination among carotenoid metabolic genes. The offset interaction between the alleles with increasing and decreasing effects on carotenoid content and the epistatic interaction among carotenoid metabolic genes were observed and these interactions complexed carotenoid profiles in breeding population. Out of the examined carotenoid metabolic genes, we confirmed by association analysis on breeding population that *PSY* and *ZEP* play a key role in controlling the carotenoid

flux in mature fruits. Consequently, a high concentration of carotenoids could be achieved by pyramiding the ideal alleles of *PSY-a* and *ZEP-e*. These important alleles were successfully genotyped using the TaqMAN-MGB SNP markers for MAS. The results obtained, along with the DNA markers used, can facilitate carotenoid metabolic engineering to potentially improve the nutritional quality of fruits.

## Supporting information

**S1 Fig. Schematic diagram to develop the trio-tagged SNPs in the alleles of 5 target genes in 13 founders.**
(TIF)

**S2 Fig.** Estimated effect size ($\zeta$) of each allele types in the 5 target genes regions for (A) $\alpha$- carotene, (B) $\beta$-carotene, (C) lutein, (D) phytoene, (E) $\zeta$-carotene, and (F) zeaxanthin, respectively. Red, blue, and purple circle indicated that the allele type among the 5 target genes had the highest effect size, the largest variance, and both the highest effect size and the largest variance, respectively. The characters in the brackets mean the product of the posterior means of $\gamma$.
(TIF)

**S3 Fig. Box plot analysis to verify the effectiveness of optimum *PSY* and *ZEP* alleles for the enrichment of β-cryptoxanthin in juice sac tissues of fruits.** Box edges represent the upper and lower quantiles, with the median value shown as a bold line in the middle of the box. The circles indicate an outlier. (A) Comparison of β-cryptoxanthin content by the number of alleles with positive effect (PSY-a) and negative effect *(PSY-c* or *-g)*. Condition 1, *PSY-a* = 0 & *(PSY-c* or *-g)* = 2 Condition 2, *PSY-a* = 0 & *(PSY-c* or *-g)* = 1; Condition 3, *PSY-a* = 0 & *(PSY-c* or *-g)* = 0; Condition 4, *PSY-a* = 1 & *(PSY-c* or *-g)* = 1; Condition 5, *PSY-a* = 1 & *(PSY-c* or *-g)* = 0; Condition 6, *PSY-a* = 2 & *(PSY-c* or *-g)* = 0. Means were tested for significant differences using Tukey's HSD test at P < 0.05, following the normality of the distribution was confirmed not to be rejected at $P = 0.05$ by Kolmogorov-Smirnov's one sample test. β-cryptoxanthin content was analyzed as square root-transformed values. N/A indicates values eliminated from the Tukey's HSD test due to insufficient data. (B) Comparison of total carotenoid content by the number of alleles with a positive effect *(ZEP-a*, *-b*, *-c*, or *-e)* under the *PSY* allele-type conditions (A). Means were tested for significant differences using Tukey's HSD test at $P < 0.05$ under conditions 2 and 5. Means between '+Allele = 1' and '+ Allele = 2' were tested for significant differences using Welch's two-sample t-test under condition 3, following the normality of the distribution was confirmed not to be rejected at $P = 0.05$ by Kolmogorov-Smirnov's one sample test. 'ns' indicates not significant at $P < 0.05$. β-cryptoxanthin content was analyzed as square root-transformed values.
(TIF)

**S4 Fig. Box plot analysis to verify the effectiveness of optimum *PSY* and *ZEP* alleles for the enrichment of violaxanthin in juice sac tissues of fruits.** Box edges represent the upper and lower quantiles, with the median value shown as a bold line in the middle of the box. The circles indicate an outlier. (A) Comparison of violaxanthin content by the number of alleles with positive effect *(PSY-a)* and negative effect *(PSY-c* or *-g)*. Condition 1, *PSY-a* = 0 & *(PSY-c* or *-g)* = 2 Condition 2, *PSY-a* = 0 & *(PSY-c* or *-g)* = 1; Condition 3, *PSY-a* = 0 & *(PSY-c* or *-g)* = 0; Condition 4, *PSY-a* = 1 & *(PSY-c* or *-g)* = 1; Condition 5, *PSY-a* = 1 & *(PSY-c* or *-g)* = 0; Condition 6, PSY-a = 2 & *(PSY-c* or *-g)* = 0. Means were tested for significant differences using Tukey's HSD test at P < 0.05, following the normality of the distribution was confirmed not to be rejected at P = 0.05 by Kolmogorov-Smirnov's one sample test. Violaxanthin contents were

analyzed as square root-transformed values. N/A indicates values eliminated from the Tukey's HSD test due to insufficient data. (B) Comparison of total carotenoid content by the number of alleles with a positive effect *(ZEP-a, -b, -c,* or *-e)* under the *PSY* allele-type conditions (A). Means were tested for significant differences using Tukey's HSD test at $P < 0.05$ under conditions 2 and 5. Means between '+Allele = 1' and '+ Allele = 2' were tested for significant differences using Welch's two-sample t-test under condition 3. ns indicates not significant at $P < 0.05$, following the normality of the distribution was confirmed not to be rejected at P = 0.05 by Kolmogorov-Smirnov's one sample test. Violaxanthin contents was analyzed as square root-transformed values.
(TIF)

**S5 Fig. Phylogenetic tree analysis of 7 *PSY* independent alleles in 13 founders by the neighbor-joining method.** The tree is divided into two major nodes. The positive allele *(PSY-a)* and negative alleles *(PSY-c* and *PSY-g)* were clustered into different nodes, revealing that these alleles might be divergent from the common ancestral *PSY*. Numbers above branches are bootstrap values based on 1000 replicates and numbers under ranches are genetic distance for each pair of nodes.
(TIF)

**S6 Fig. Phylogenetic tree analysis of 11 *ZEP* independent alleles in 13 founders by the neighbor-joining method.** The tree is divided into two major nodes. The positive alleles *(ZEP-a*, *b, c, e)* and negative alleles *(ZEP-f, g, h, k)* are mixed in those nodes, revealing that the functional divergence of those alleles might have occurred after the divergence of 2 functionally different *ZEP* ancestral alleles. Numbers above branches are bootstrap values based on 1000 replicates and numbers under ranches are genetic distance for each pair of nodes.
(TIF)

**S1 Table. Information on reliable trio-tagged SNPs with lineage by trio-analysis.**
(PDF)

**S2 Table. SNP genotype of trio-tagged SNPs for 5 target genes.**
(PDF)

**S3 Table. Allelic genotype of 5 target genes in major varieties and cultivars.**
(PDF)

## Author Contributions

**Conceptualization:** Hiroshi Fujii.

**Data curation:** Hiroshi Fujii, Aiko Sugiyama, Mitsuo Omura.

**Funding acquisition:** Hiroshi Fujii, Takehiko Shimada.

**Investigation:** Keisuke Nonaka, Tomoko Endo, Mitsuo Omura, Takehiko Shimada.

**Resources:** Keisuke Nonaka.

**Software:** Hiroshi Fujii.

**Validation:** Mai F. Minamikawa, Kosuke Hamazaki, Hiroyoshi Iwata.

**Writing – original draft:** Takehiko Shimada.

**Writing – review & editing:** Takehiko Shimada.

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
