## [Decision Letter · Decision Letter 0]

10 Dec 2020

PONE-D-20-34918

Allelic composition of carotenoid metabolic genes in 13 founders influences carotenoid composition in fruits among Japanese citrus breeding population

PLOS ONE

Dear Dr. Shimada,

Thank you for submitting your manuscript to PLOS ONE. After careful consideration, we feel that it has merit but does not fully meet PLOS ONE’s publication criteria as it currently stands. Therefore, we invite you to submit a revised version of the manuscript that addresses the points raised during the review process.

We look forward to receiving your revised manuscript.

Kind regards,

Chunxian Chen, Ph.D.

Academic Editor

PLOS ONE

Journal Requirements:

"This work was partially supported by a grant from the Ministry of Agriculture,

Forestry, and Fisheries of Japan (Genomics-based Technology for Agricultural

Improvement, HOR-2003, DNA-marker breeding project), by a grant from the Project

of the Bio-oriented Technology Research Advancement Institution, NARO (the special

scheme project on advanced research and development for next-generation technology),

and by a Grant-in-Aid for Scientific Research of Japan Society for the Promotion of

Science (JSPS), Grant Number 23580055.".

i) We note that you have provided funding information that is not currently declared in your Funding Statement. However, funding information should not appear in the Acknowledgments section or other areas of your manuscript. We will only publish funding information present in the Funding Statement section of the online submission form.

ii) Please remove any funding-related text from the manuscript and let us know how you would like to update your Funding Statement. Currently, your Funding Statement reads as follows:

"The author(s) received no specific funding for this work.".

 iii) Please include your amended statements within your cover letter; we will change the online submission form on your behalf.

Additional Editor Comments (if provided):

The paper was to identify alleles of carotenoid biosynthetic genes using citrus breeding parents and populations using the SureSelect target enrichment method. The results are of great value for understanding the diversity of carotenoid gene expressions and compositions. The paper was well written. The authors need attentions to some editorial issues in the paper.

Introduction:

Some early relevant works on citrus carotenoid biosynthetic genes/alleles and the pigment compositions should be cited, for example, those from Ollitrault’s group (e.g., Fanciullino et al. 2007) and Dr. Gmitter’s group (e.g., Chen et al. 2010).

Abstract and main text:

Abbreviations are needed only when they are used afterwards (no need of abbreviations otherwise). This rule is also applied to the main text.

Figures and tables:

They should be understood well without reference to the main text. Some legends and captions should have a bit more illustration. Likewise, abbreviations should be redefined provided that they are used there afterwards.

Reviewers' comments:

Reviewer's Responses to Questions

**Comments to the Author**

1. Is the manuscript technically sound, and do the data support the conclusions?

Reviewer #1: Yes

Reviewer #2: Yes

2. Has the statistical analysis been performed appropriately and rigorously? 

Reviewer #1: Yes

Reviewer #2: I Don't Know

3. Have the authors made all data underlying the findings in their manuscript fully available?

Reviewer #1: Yes

Reviewer #2: Yes

4. Is the manuscript presented in an intelligible fashion and written in standard English?

Reviewer #1: Yes

Reviewer #2: Yes

5. Review Comments to the Author

Reviewer #1: “Carotenoids are C40 lipophilic isoprenoid pigments biosynthesized from 5-carbon isoprene units” should be “Carotenoids are mostly C40 lipophilic isoprenoid pigments biosynthesized from 5-carbon isoprene units”. There are C30 carotenoids in citrus.

Valencia orange (Citrus sinensis Osbeck) and Lisbon lemon (Citrus limon Burm. f.) should be Valencia orange (C. sinensis Osbeck) and Lisbon lemon (C. limon Burm. f.)。

For a better understanding, the procedures in “SureSelect target enrichment of carotenoid metabolic genes in 13 founders” are suggested to be present with a diagram.

The abbreviations in the text, such as “PHY, 400 nm for ZCA, 452 nm for t-VIO, c-VIO, LUT, BCR, ACA, and ZEA, and 453 nm for BCA”, make it difficult to read and understand, full terms are suggested to be present.

I guess the sentences below Figure titles are notes, but not the text, so please add “Notes” before them. However, the ‘Fig. 1. Carotenoid metabolic pathway in citrus fruits’ should be better put in the section of ‘Introduction’, not in the ‘Results’. And QTL and eQTL sites in Fig.1 should be mentioned as specially generated from juice sacs, but not from the flavedo, albedo and segment membrane, for carotenoids and carotenogenesis in the later fruit tissues are quite different.

Please specify those negative, positive and optimum alleles were based on previous Estimated effect size from Bayesian statistical analysis.

Please rephrase the sentence of ‘When the mean values of conditions 1 and 2 were compared with those of condition 3, it was clear that PSY-c and PSY-g had negative effects on reducing the carotenoid concentration.’

Reviewer #2: 1. According to introduction, the authors mentioned “eQTLs of PSY, HYb, and ZEP could be mapped on the loci of their corresponding genes, revealing that their transcription is regulated primarily by cis-elements in their promoter regions.” So what are the polymorphism in the promoter region of these genes?

2. Filtering parameters for the SNP should be added. In the text, the authors mentioned that ”The number of SNPs in PSY, HYb, ZEP, and NCED with more than readable SNP quality score (>150) according to manufacturer’s description were 107, 31, 54 and 19 respectively”, and “Based on the SNP information, 17 SNP markers for PSY, 15 for HYb, 31 for ZEP, 8 NCED, and 5 for TCL were developed for a SNP genotyping assay using the GoldenGate assay system (Illumina) and Fluidigm BioMark™ HD assay system (Fluidigm).” in page 17. What kind of SNPs used? Nonsynomous mutations in coding regions or others?

3. Figure 9, the normal distribution and variance should be assessed before the Welch's test and the Tukey's HSD test were used for significance analysis, and the information should be added.

4. A mistake in page 7 “PYS, HYb, ZEP, and TCL possessed 7, 11, 5, and 4 independent alleles in 13 founders”, PYS should be PSY.

6. PLOS authors have the option to publish the peer review history of their article (what does this mean?). If published, this will include your full peer review and any attached files.

Reviewer #1: **Yes: **Dr. Juan Xu

Reviewer #2: No

---

## [Author Response · Author response to Decision Letter 0]

11 Jan 2021

Dear Reviewers

We are appreciated with careful checks, the meaningful comments and suggestions of reviewer, which are helpful to improve the quality of our manuscript. We describe our responses against each comment in below. All modifications in new manuscript are written in red characters. 

Reviewer #1:

 "Carotenoids are C40 lipophilic isoprenoid pigments biosynthesized from 5-carbon isoprene units" should be "Carotenoids are mostly C40 lipophilic isoprenoid pigments biosynthesized from 5-carbon isoprene units". There are C30 carotenoids in citrus.

Answer) The information of C30 apocarotenoids is added in ‘Introduction’

Valencia orange (Citrus sinensis Osbeck) and Lisbon lemon (Citrus limon Burm. f.) should be Valencia orange (C. sinensis Osbeck) and Lisbon lemon (C. limon Burm. f.)。

Answer) Academic names of them are changed according to reviewer’s comment. 

For a better understanding, the procedures in "SureSelect target enrichment of carotenoid metabolic genes in 13 founders" are suggested to be present with a diagram.

Answer) According to reviewer’s comment, a new diagram is constructed and is added as a new Fig. S1 in new manuscript.

The abbreviations in the text, such as "PHY, 400 nm for ZCA, 452 nm for t-VIO, c-VIO, LUT, BCR, ACA, and ZEA, and 453 nm for BCA", make it difficult to read and understand, full terms are suggested to be present.

Answer) Full compound names are described in new manuscript.

I guess the sentences below Figure titles are notes, but not the text, so please add "Notes" before them. However, the 'Fig. 1. 

Answer) ‘Note’ is added below the title of figure legends

Carotenoid metabolic pathway in citrus fruits' should be better put in the section of 'Introduction', not in the 'Results'. And QTL and eQTL sites in Fig.1 should be mentioned as specially generated from juice sacs, but not from the flavedo, albedo and segment membrane, for carotenoids and carotenogenesis in the later fruit tissues are quite different.

Answer) Fig. 1 is moved to ‘Introduction’ according to reviewer’s comment. In addition, the information which sample was used in fruit tissues was lacked in the previous manuscript. We used juice sac tissues in the analysis. This information is added at various places in new manuscript. 

Please specify those negative, positive and optimum alleles were based on previous Estimated effect size from Bayesian statistical analysis.

Answer) We defied the allele with more than one third of the max effect size (ζ) based on results of Fig.7 as positive or negative alleles for the measured carotenoids. For an example in total carotenoid, the alleles having more than (ζ =) + 0.249 or – 0249 effect size are defined as the ‘positive’ or the ‘negative’ effect alleles.

 On one hand, the optimum allele means to possess the strongest effect among positive alleles, that is PSY-a and ZEP-e for total carotenoid. 

 These explanations are added separately in the result of new manuscript.

Please rephrase the sentence of 'When the mean values of conditions 1 and 2 were compared with those of condition 3, it was clear that PSY-c and PSY-g had negative effects on reducing the carotenoid concentration.'

Answer) The precious sentence was not correct as reviewer suggested. This sentence is modified in new manuscript. In addition, a new sentence is added to describe allelic interaction between positive and negative alleles, which was observed in in condition 4.

Reviewer #2: 

1. According to introduction, the authors mentioned "eQTLs of PSY, HYb, and ZEP could be mapped on the loci of their corresponding genes, revealing that their transcription is regulated primarily by cis-elements in their promoter regions." So what are the polymorphism in the promoter region of these genes?

Answer) Citrus varieties possess high heterozygosity and high polymorphism in their genomes. Promoter sequences of citrus varieties are highly varied and numerous polymorphisms could be detected when compared. We considered that some or most of those polymorphisms might be meaningless in contrast to homogenous plants with less polymorphism. Sequence variation in promoter region would like to occur highly by the combination of insertion, deletion and point mutation among citrus varieties, some of which might happen in the process of vegetative reproduction for a long time period. On one hand, functional cis-elements in promoter region should be conserved within the functionally synonymous alleles. However, comparing analysis of promoter sequence among citrus varieties cannot provide the definitive information to find the cis-element site regulating for the transcription level from the numerous polymorphisms. From these reasons, we carried out allele mining method using SNP in gene region. We add the reason why SNP is collected from gene region in ‘Result’ of new manuscript. 

2. Filtering parameters for the SNP should be added. In the text, the authors mentioned that "The number of SNPs in PSY, HYb, ZEP, and NCED with more than readable SNP quality score (>150) according to manufacturer's description were 107, 31, 54 and 19 respectively", and "Based on the SNP information, 17 SNP markers for PSY, 15 for HYb, 31 for ZEP, 8 NCED, and 5 for TCL were developed for a SNP genotyping assay using the GoldenGate assay system (Illumina) and Fluidigm BioMark™ HD assay system (Fluidigm)." in page 17. What kind of SNPs used? Nonsynomous mutations in coding regions or others?

Answer) For better understanding of filtering of SNPs, new S1 Fig. is constructed. The condition of MARCO software is also added in new manuscript. 

 Regarding for the question of what kind of SNPs, we used all inheritable SNPs in gene region among all examined trio-sets (parents and their children), in which both of nonsynomous and synonymous mutation were included. To supplement this information, S1 Table is modified to add the information of amino acid information on SNP position.

3. Figure 9, the normal distribution and variance should be assessed before the Welch's test and the Tukey's HSD test were used for significance analysis, and the information should be added.

Answer) We confirmed the normality of the distribution by Kolmogorov-Smirnov’s one sample test before the significance analysis. This explanation was added in notes of Figure. 9, S3 Fig and S4 Fig.

4. A mistake in page 7 "PYS, HYb, ZEP, and TCL possessed 7, 11, 5, and 4 independent alleles in 13 founders", PYS should be PSY.

Answer) PYS is modified to PSY.

---

## [Editor Report · Decision Letter 1]

20 Jan 2021

Allelic composition of carotenoid metabolic genes in 13 founders influences carotenoid composition in juice sac tissues of fruits among Japanese citrus breeding population

PONE-D-20-34918R1

Dear Dr. Shimada,

We’re pleased to inform you that your manuscript has been judged scientifically suitable for publication and will be formally accepted for publication once it meets all outstanding technical requirements.

Kind regards,

Chunxian Chen, Ph.D.

Academic Editor

PLOS ONE
---

## [Editor Report · Acceptance letter]

22 Jan 2021

PONE-D-20-34918R1 

Allelic composition of carotenoid metabolic genes in 13 founders influences carotenoid composition in juice sac tissues of fruits among Japanese citrus breeding population 

Dear Dr. Shimada:

I'm pleased to inform you that your manuscript has been deemed suitable for publication in PLOS ONE. Congratulations! Your manuscript is now with our production department. 

Kind regards, 

on behalf of

Dr. Chunxian Chen 

Academic Editor

PLOS ONE